# How to choose a design thinking method for teaching the design of localization: A two-dimension linguistic fuzzy model with two-tuples

You-Lei Fu[1,2], Linxin Zheng[1]*, Kuei-Chia Liang[3]*, Ruoqi Dai[4]

1 School of Design and Fashion, Zhejiang University of Science and Technology, Hangzhou, Zhejiang, China, 2 Anji-ZUST Research Institute, Huzhou, Zhejiang, China, 3 Department of Design, National Taiwan Normal University, Taipei, Taiwan, 4 FILA Sports Co., Ltd, Xiamen, Fujian, China

* zhenglinxin@zust.edu.cn (LZ); kcliang@ntnu.edu.tw (KCL)

**Data Availability Statement:** All relevant data are within the manuscript and its Supporting information files, and upload the data to the

## Abstract

There are many different types of scientific design thinking methods, but it is necessary to evaluate the applicability of the methods to the components of the design teaching curriculum in universities. Therefore, this study assesses the applicability of design thinking in terms of "design practice" and "locality" based on the local design education philosophy and the characteristics of the students and courses. A two-dimensional linguistic fuzzy model with two-tuples was proposed, and the assessment values of 36 experts were statistically analysed using the Delphi, triangular fuzzy number, Euclidean distance, two-dimension linguistic label (2DLL), and two-dimensional linguistic weighted arithmetic aggregation (2DLWAA) methods. The results highlighted the 12 categories of design thinking methods that are most applicable to teaching and learning, indicating the basic views of university design faculty on the application of design thinking methods. Finally, the new design teaching methods have been validated and constructed through years of teaching practice, and have some reference value for teaching design courses in universities.

## 1 Introduction

### 1.1 Design thinking and teaching

Design thinking is one of the most relevant areas of design and innovation [1]. The application of user-centred design thinking methods for researching data, synthesizing and summarizing information, and communicating results and design connotations is very common, and these methods and techniques produce a variety of interactions that are integral to the process [2, 3]. A study by Rauth et al. [4] found that design thinking education allows for different levels of creative knowledge, skills, and thinking patterns, culminating in an ability known as "creative confidence". Based on these results, design education demonstrates how the process contributes to the development and understanding of design creativity and therefore defines "design thinking" as a model of learning geared towards creative confidence.

figshare repository, https://doi.org/10.6084/m9.figshare.25010591.

**Funding:** This research is supported by "Exploration and Practice of Diversified Collaborative Education Model in Creative Design Programs in the Context of New Humanities and Social Sciences Background (2021090048)", Research and Reform Practice Project of New Humanities and Social Sciences by the Ministry of Education, 2021.

**Competing interests:** The authors have declared that no competing interests exist.

**Abbreviations:** 2DLL, 2-dimension linguistic label; 2DL-LIA, 2-dimension linguistic lattice implication algebra; 2DLM, 2-dimension linguistic fuzzy model; 2DLV, 2-dimension linguistic Value; 2DLWAA, 2-dimension linguistic weighted arithmetic aggregation; DT-2DL-LIA, 2-dimension linguistic lattice implication algebra of design teaching; DT-2DLV, 2-dimension linguistic value of design teaching; LIA, Lattice implication algebra.

Design thinking focuses on developing the ability to think creatively and confidently, and to be self-innovative. In terms of the purpose of learning about this process, students can become action-oriented thinkers who employ it primarily for personal usage, or proactive advocates who further extend it for use by others and groups. In the Asia-Pacific region, China, Korea, and India have promoted design thinking in university education by establishing programmes focused on fostering design thinking [5]. Design thinking has been praised by a growing number of scholars and educational researchers for its potential to improve curriculums and pedagogy [6]. Many design and management schools in North America and elsewhere now include design thinking courses. Australia offers the opportunity to undertake courses using design thinking practices. Swinburne University is developing design thinking courses in both Melbourne and Hong Kong [7]. Wooff [5] suggests that education should focus more on the design dimension of humanistic features and include design thinking as an integral part of education. Design thinking seeks to utilize knowledge and practice to find workable solutions to meet people's needs and interests in the context of contemporary social challenges. A human-centred approach to the process also promotes empathy and contributes to students' character development [8]. Design thinking thrives on ambiguity and uncertainty, and as such, it broadens students' educational experiences by encouraging creative and reflective thinking, self-awareness, and social awareness. In short, the design thinking approach fosters many of the desirable characteristics identified as 21st-century competencies [9].

If design thinking is applied to the teaching and practice of design, teachers can strengthen students' thinking and promote collective thinking in the class as a whole through control of the method management and process. They can combine the visual design thinking process with logical thinking methods, finally teaching the students to stop only looking for answers from relevant cases, to avoid the situation of transitional borrowing and similarity. Not all design thinking methods are suitable for design teaching (DT) course features, and there is a need to analyse the suitability of design thinking methods combined with teaching. Furthermore, no one method can be used alone and needs to be adopted collectively as needed in a given situation [10]. Thus, according to the school's preferences and student characteristics, teachers can be organized into focus group discussions to filter different methods, so similar ones can be integrated or inapplicable ones removed.

## 1.2 Locality and design teaching

The shortcomings of the globalization advocated by the West have led to the widespread attention and importance of "localism". This is a reorganization reaction to the constant changes of modernization, urbanization, globalization, and symbolic consumerization, and the reorganization of local characteristics that have been neglected by the strong Western modernization, such as the climate and cultural identity, in addition to materials, community characteristics (i.e., geography, population, and social development), etc. [11]. The scale concept of locality is a boundary range with relativity, from cultural regions, peoples, and nations, to specific administrative geographic areas, as well as villages, communities, or streets [12, 13]. The geographical design has an important and beneficial impact on local target users, increasing attractiveness, satisfaction, trust, and productivity [14].

The teaching practice of "localism" as a design concept is based on the characteristics of the times and the natural and human conditions of the region, and advocates the integration of design with the region, including with the culture, environment, behaviour, industry, and materials. Pragmatic approaches emphasize close interaction with local practices and users to facilitate the practical usability of the proposed course [15]. The integration of design thinking tools such as workshops, concept boards, prototypes, and experiments is also

suitable for incorporation into topic-specific design courses and contributes to pedagogical effectiveness [16].

Based on the six stages of design thinking, this study incorporates six processes in the curriculum of regional design teaching, such as theoretical explanation, research and observation, the definition of ideas, conceptual exploration, scheme design, and design discussion. Students are required to conduct various forms of assignment assessment such as data collection and analysis, design research, class and panel discussion, and design appreciation. The new assessment model changes the previous practice of measuring students' learning effectiveness only by midterm or final exam results and combines various evaluations such as classroom performance, assignment quality, participation in competitions, or the social value embodiment of actual projects to assess students' knowledge mastery and application ability. However, even so, the teaching method still focuses more on students' learning and discussion of existing knowledge, while students' tasks are assigned in a more open way, without limiting the use of specific methods and expressions, but only regulating the design process and assignment content.

This research aims to incorporate design thinking into the design education process in universities and to develop a pedagogical approach to design courses on the theme of "localism". However, existing thinking techniques are extensively utilized in management, creative, engineering, and other sectors, and there are many various types of approaches. It is therefore necessary to investigate and describe the suitable teaching methods in regional design education.

## 1.3 Selection of teaching methods

Teaching methods provide an explicit system that helps teachers understand and impart knowledge and guides them through the process of learning activities [17]. Design courses have often used two models in the past; one is a theory-oriented case presentation model where knowledge and skills are imparted through lectures, reading, and viewing (listening) [18]. The other is a practice-oriented project-based model [19], in which students are motivated to actively learn knowledge and skills by engaging in exploration, discussion, and experimentation [20]. However, although design courses often introduce a variety of conceptual design approaches [21], there is a general lack of consensus on the choice of specific pedagogical approaches and insufficient research on how to select materials, pedagogical approaches, and learning objectives for design courses.

Several studies have argued that teaching should favour a practice-oriented model in which teachers become coaches and developers who facilitate individual learning through engagement with group members to solve complex social problems [22, 23]. Therefore, the choice of pedagogical methodology becomes crucial in influencing the effectiveness of learning. Goodyear [24] reviews and analyses several relevant work examples from the perspectives of design epistemology, phenomenology, and praxis to explore ways of building design capacity within higher education institutions. Marjanovic [25] employs the action design research (ADR) method to design and evaluate a new pedagogical framework. Additionally, assessment methods through scale tools have also been applied, such as Scott's Course Experience Questionnaire (CEQ) analysis [26], and Bernstein et al.'s [27] strategic approach to building a design curriculum using multicriteria student assessments and ranking their respective knowledge utilizing Likert scales. Wu and Chen [28] conducted a more in-depth exploration of assessment methods, applying latent semantic analysis, the Delphi method, and a combination of the revised Bloom's taxonomy and hierarchical analysis, and prioritized four instructional methods based on the results.

Evaluation information in the form of language is not directly involved in mathematical operations, and decision-making on it requires multiple experts to form a group decision, which then requires information transformation and aggregation of the information of each program, followed by sorting, comparison, and selection of the best. At present, the methods of processing linguistic evaluation information are mainly divided into three categories: one is the analytical method of evaluation information transformation, which transforms linguistic evaluation information into fuzzy numbers, and then carries out arithmetic operations and analysis, such as the fuzzy analytic hierarchy process (F-AHP) [29], the second is the symbol transfer-based analysis method, which is based on the calculation of linguistic evaluation information values directly from the set of linguistic evaluation information, such as RANking COMparison (RANCOM) [30], the third category of methods is the 2-tuples linguistic analysis method [31], which aims to transform the linguistic evaluation information given by the decision maker into 2-tuples linguistic symbols, and then use the 2-tuples linguistic algorithm and the agglomerative operator to perform information agglomerative analysis. By comparing and analyzing the above methods, it is found that the 2-tuples linguistic analysis method has great superiority in dealing with model and information loss, which makes the calculation results of linguistic evaluation information more accurate. In summary, the methods for assessing the applicability of teaching methods are one-dimensional results from literature or statistical analysis and do not include the thematic characteristics of the course as a second dimension, so the assessment of suitability is not comprehensive enough. For this reason, this study proposes a two-dimensional expert assessment method, setting up two concepts involving the thematic character of the course and pedagogical mode, and adopting a two-dimensional linguistic fuzzy model with two-tuples to incorporate mixed statistical methods, including the Delphi, triangular fuzzy number, Euclidean distance, two-dimension linguistic label (2DLL), and two-dimensional linguistic weighted arithmetic aggregation (2DLWAA) methods.

## 2 Research methods and procedures

This study proposes an assessment and curricular application of the pedagogical applicability of design thinking approaches, and the specific methodology is depicted in Fig 1. The first phase comprises creating a design thinking method selection questionnaire, followed by an expert assessment involving the Delphi technique to determine the categories of design thinking methods employed in design education D. The second stage establishes the triangle fuzzy functions and linguistic variables that will be utilized to determine the instruments to be evaluated, as well as the two-dimensional linguistic model (2DLM) with the two-dimensional ordered sets ($S$ and $N$). The third step is to set the δ values of the two dimensions, and to obtain the weight assessment values of "design practice" and "locality" dimensions in design teaching by constructing a questionnaire. The fourth step is the assessment of the pedagogical adaptability of the design thinking method, which first defines the assessment model $R_{app}$, then constructs a questionnaire to obtain the experts' applicability assessment values, converts the assessment value d into a triangular fuzzy number R, obtains the solution triangular fuzzy number, and finally aggregates the two-dimensional linguistic value (2DLV) of each expert with the 2DLWAA aggregation algorithm to obtain the two-dimensional linguistic fuzzy model with two-tuples. The fifth step uses the two-dimensional binary fuzzy semantic evaluation values and δ values to determine the comparative relationship graph of design thinking methods; finally, the design thinking methods suitable for teaching are selected and applied and validated through the teaching of environmental retrofit design courses. If a suitable method is not obtained, it is possible to go back to the first step to revise the defined pedagogical method category.

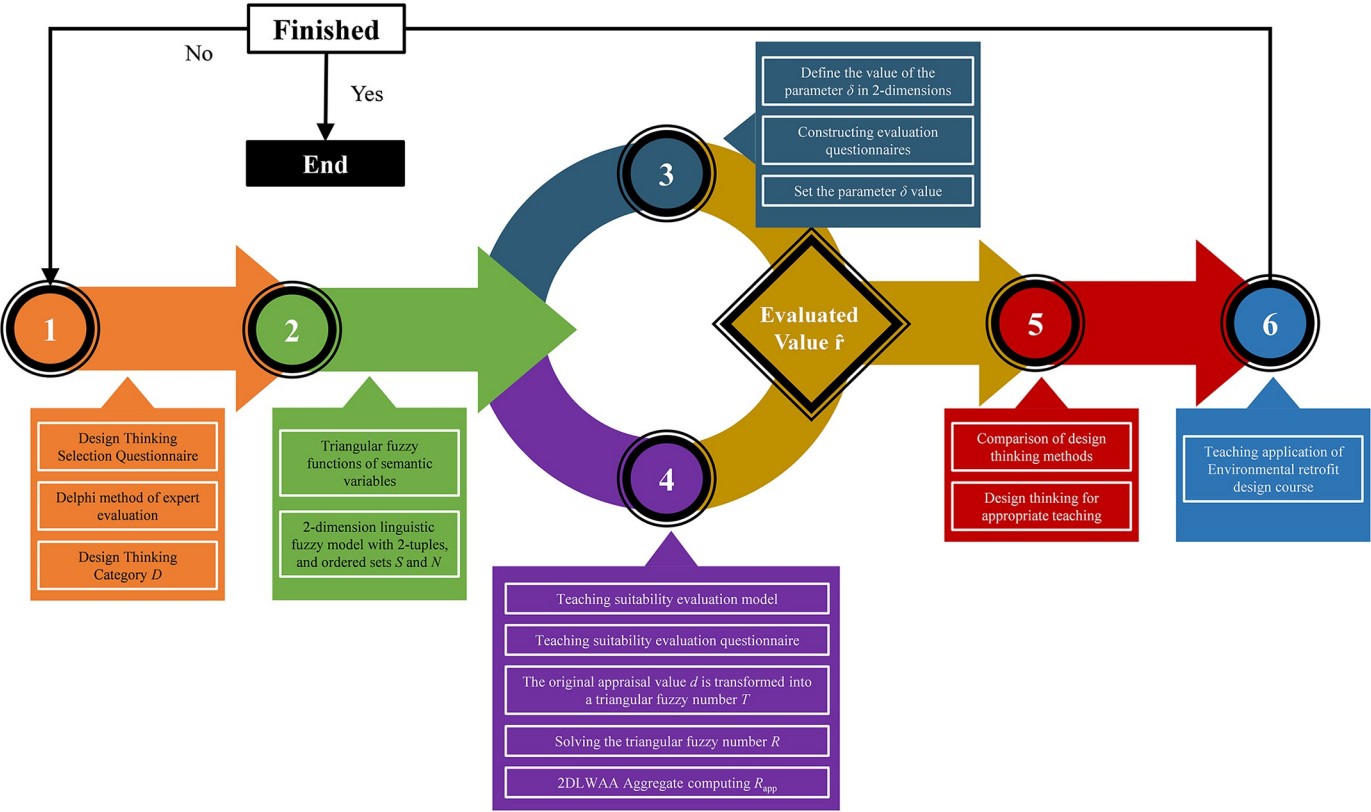

**Fig 1. Research process.**

## 2.1 Fuzzy Delphi method

**2.1.1 Expert evaluation.** The Delphi technique is a research method that employs a scientific approach to bring together the consistent opinions of experts and scholars on a specific topic or event. The Delphi method is often utilized to construct criteria that must be cohesive and highly credible or scales that are used as assessment tools [32, 33]. To compensate for the shortcomings of the traditional Delphi method, the combination of the fuzzy Delphi method was proposed [34], which has been widely applied in the fields of quality analysis, performance evaluation, economic analysis, and industrial strategy for many years [35, 36], and has been proven to be reliable and effective. Usually, the number of experts invited by the fuzzy Delphi method is between 10 and 15 [37–39]. The experts should be selected based on their representative professional background and high level of expertise in the subject matter of the opinion consultation. The expert group for this study consisted of 41 members of the design faculties (environmental, product, and visual communication design) from more than 10 universities in China, Taiwan, and Korea. Five experts participated in the selection of teaching methods and 36 in the assessment to analyse and verify the applicability of the selected methods in the dimensions of "design practice" and "locality".

**2.1.2 Informed consent of the participants.** The recruitment of assessment experts for this analysis took place on 10–20 June 2022 for the assessment of instructional methods and on 23 June 2022 for the evaluation of the applicability of instructional methods. The evaluation utilized expert panel interviews and web-based questionnaires to obtain subjective assessment data. Before data collection, the respondents of the main content of the study were informed

through verbal notification that this assessment was anonymous and that it would be used only for research purposes, and their consent to begin the assessment measurements was obtained. In addition, the same instructions were given in the web-based questionnaire, and consent was acquired to begin the assessment measurements.

**2.1.3 Triangular fuzzy function.** Traditional quantitative representations of fuzziness or uncertainty are difficult to express and quantify accurately, hence the idea of linguistic variables is applicable [40, 41]. Based on the study literature of Kahraman et al. [43], fuzzy numbers were utilized to determine the appraised value of the expert questionnaire [42] and applied in the Delphi technique using mathematical operations.

Let $\tilde{A}$ be a fuzzy set in X, $\mu_{\tilde{A}}(x)$: x→[0,1]. As x is closer to 1, this means that the set contains more elements of x. This value is called the degree of membership and $\mu_{\tilde{A}}(x)$ is called the membership function [43].

Natural semantic variables are the most realistic and direct description of information [44, 45], and the applicability of this study was assessed using a seven-grade scale, as shown in Fig 2.

**2.1.4 Conversion of triangular fuzzy numbers to explicit values.** The affiliation function $T_{ij}$ of the triangular fuzzy number is assumed to be of the triangular type, $T_{ij} = (a_{ij}, b_{ij}, c_{ij})$. We choose the arithmetic mean as an aggregation algorithm to obtain the group assessment value for each questioned term (Eq 1).

$$T_{ij} = \left(\frac{1}{m}\sum_{i=1}^{m} a_{ij}, \frac{1}{m}\sum_{i=1}^{m} b_{ij}, \frac{1}{m}\sum_{i=1}^{m} c_{ij}\right) \tag{1}$$

However, these group assessment values are triangular fuzzy number sets that do not exactly correspond to any semantic numbers and therefore require defuzzy number operations. Although many distance measurement function algorithms can be used [46], the present study utilizes the Euclidean distance method to approximate the fuzzy semantic ones. This method is an efficient and simple way to calculate the distance between two triangular fuzzy numbers [47–50].

With two triangular fuzzy numbers $T_1 = (a_1, b_1, c_1)$ and T2 = $(a_2, b_2, c_2)$, the operation Eq 2 for the distance d($T_1$, $T_2$) between these two fuzzy numbers is as follows:

$$d(T_1, T_2) = \sqrt{P_1(a_1 - a_2)^2 + P_2(b_1 - b_2)^2 + P_3(c_1 - c_2)^2} \tag{2}$$

In the formula, $P_1$, $P_2$, and $P_3$ denote the weights of the parameters ($a_i$, $b_i$, $c_i$) measuring the fuzzy number $T_i$, $P_i \in [0,1]$, $\sum P_i = 1$. In this study, the P values are set as $P_1 = 0.2$, $P_2 = 0.6$, and

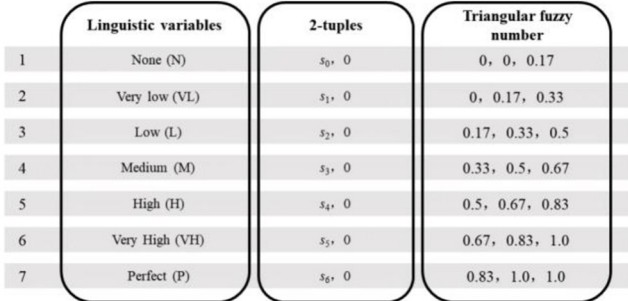

| | Linguistic variables | 2-tuples | Triangular fuzzy number |
|---|---|---|---|
| 1 | None (N) | $s_0$, 0 | 0, 0, 0.17 |
| 2 | Very low (VL) | $s_1$, 0 | 0, 0.17, 0.33 |
| 3 | Low (L) | $s_2$, 0 | 0.17, 0.33, 0.5 |
| 4 | Medium (M) | $s_3$, 0 | 0.33, 0.5, 0.67 |
| 5 | High (H) | $s_4$, 0 | 0.5, 0.67, 0.83 |
| 6 | Very High (VH) | $s_5$, 0 | 0.67, 0.83, 1.0 |
| 7 | Perfect (P) | $s_6$, 0 | 0.83, 1.0, 1.0 |

**Fig 2. Fuzzy affiliation function of semantic variables.**

$P_3 = 0.2$, because the parameter bi is the most representative of the triangular fuzzy number function, while ai and ci are relatively low. Finally, using the distance formula, the triangular fuzzy number can be defuzzified (Eq 3).

$$R = \frac{d^-}{d^- + d^*} \tag{3}$$

$R$ in Eq 3 denotes the value of defuzzification, and the fuzzy number of the best-evaluated value is defined as $T^* = (1, 1, 1)$, and the fuzzy number of the worst-evaluated value is defined as $T^- = (0, 0, 0)$. $d^* = (T, T^*)$ denotes the distance between the fuzzy number $T$ and the best fuzzy number, and $d^- = (T, T^-)$ denotes the distance between the fuzzy number $T$ and the worst fuzzy number.

## 2.2 The two-dimensional linguistic fuzzy model with two-tuples

**2.2.1 Preliminaries.** Herrera and Martínez developed a linguistic fuzzy model represented by two-tuple parameters [51], which effectively overcomes the problem of loss of detailed information in the process of linguistic information processing caused by approximation methods in performing decision analysis and makes the calculation of semantic evaluation information more accurate. The two-tuple linguistic fuzzy model represents a pair of values $(s, \alpha)$ as linguistic variable information, where s is the representative symbol of the fuzzy linguistic variable and $\alpha$ represents the value of symbolic translation [52]. Inspired by the linguistic two-tuples, a method to extend one-dimensional evaluation to two dimensions was proposed [53–56]. Using two two-tuples to represent the two-dimensional linguistic value (2DLV), denoted as $\hat{r} = (s_{i1}, h_{i2})$, let R = {$\hat{r}_1, \hat{r}_2, \ldots, \hat{r}_n$} be the set of two 2DLVs of a two-tuple model. To solve the problem of comparability and incomparability of 2DLVs, a two-dimensional linguistic lattice implication algebra (2DL-LIA) is constructed by a direct product of two linguistic tag sets with an isomorphic mapping of the lattice implication algebra (LIA). The linguistic evaluation set is extended to a lattice structure, which is equivalent to a binary array of values.

The linguistic evaluation set $S = \{s_i | i = 1, \ldots, g\} = \{s_0:N, s_1:VL, s_2:L, s_3:M, s_4:H, s_5:VH, s_6:P\}$, and the symbol si denotes the evaluation value of a linguistic variable. The total number of variables is $|S| = g+1$ of S. This ordered set is usually defined as $\forall s_i, s_j \in S. s_i \leq s_j \Leftrightarrow i \leq j$. H denotes another set with a linguistic evaluation with the same characteristics as S, and H is the total number of variables $|H| = t+1$. The range of fuzzy linguistic variables used in this study is shown in Fig 3.

The numerical value $\beta_s$ denotes the number of granularity interval in the linguistic evaluation set S and $\beta_s \in [0, g]$, so that $(s_{i1}, \alpha_{i1})$ is the two-tuple linguistic form corresponding to $\beta$ and is expressed as a function $\Delta_1$, as detailed in Eqs 4 and 5.

$$\Delta_1 : [0, g] \rightarrow S \times [-0.5, 0.5) \tag{4}$$

$$\Delta_1(\beta_s) = (s_{i1}, \alpha_{i1}) = \begin{cases} s_{i1}, & i1 = Round(\beta_s) \\ \alpha_{i1} = \beta_s - i1, & \alpha_{i1} \in [-0.5, 0.5) \end{cases} \tag{5}$$

The numerical value $\beta_h$ denotes the number of granularity interval in the linguistic evaluation set $H$ and $\beta_h \in [0, t]$, so that $(h_{i2}, \alpha_{i2})$ is the two-tuple linguistic form corresponding to $\beta$

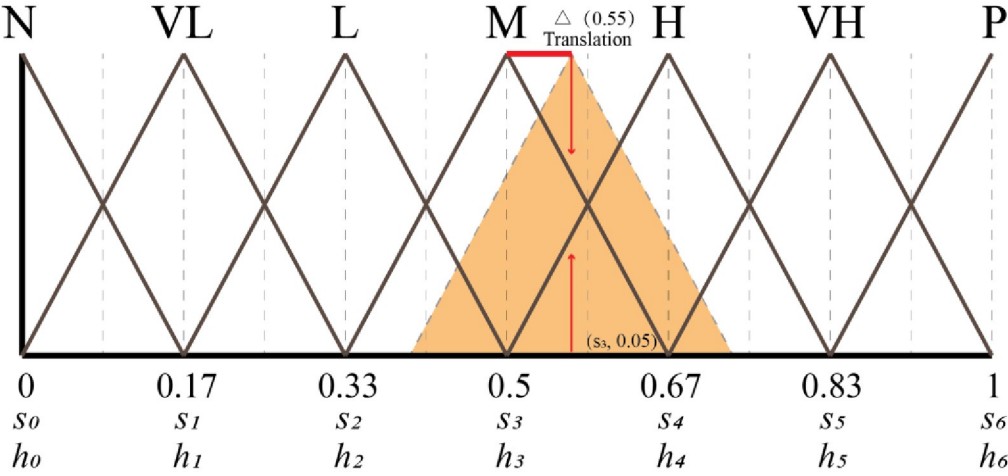

**Fig 3. Example of triangular affiliation function and 2-tuples linguistic representation of fuzzy linguistic variables.**

and is expressed as a function $\Delta_2$, as detailed in Eqs 6 and 7.

$$\Delta_2 : [0, t] \rightarrow H \times [-0.5, 0.5)$$ (6)

$$\Delta_2(\beta_h) = (h_{i2}, \alpha_{i2}) = \begin{cases} h_{i2}, & i2 = Round(\beta_h) \\ \alpha_{i2} = \beta_h - i2, & \alpha_{i2} \in [-0.5, 0.5) \end{cases}$$ (7)

Round denotes the "rounding" operation and $\alpha$ denotes the distance between the numerical value $\beta$ and the index $i$. As shown in Fig 5, an evaluation value $\beta_s = 0.55$, $i1 = 3$, $\alpha_1 = 0.55-0.5 = 0.05$, $\Delta_1(\beta_s)$ is transformed into a two-tuple fuzzy linguistic $(s_3, 0.05)$, which means it is 0.05 higher than the option "Medium (M)".

**2.2.2 Conversion of a two-tuple fuzzy linguistic number.** In this study, fuzzy numbers are integrated into binary fuzzy semantics [56]. Given an interval $T = [a, b]$, $c \in [a, b]$, set sk is a semantic variable symbol in the ordered semantic set of $S = \{s_i| i = 1,..., g\}$ the symbols of semantic variables in the ordered semantic set, where $k$ denotes the position in the semantic set and the explicit value of the semantic variable $s_k$ is $v$. Thus, $\alpha \in [-t, t] = [-0.083, 0.083]$, $t = 1/2g = (b-a)/2 \approx 0.083$. The function that converts the fuzzy semantic numbers to values is denoted as $\Delta:T \rightarrow S$, $\Delta':S \rightarrow T$, with the interval range $T = [0, 1]$, as detailed in the transformation Eq 8:

$$\Delta'((s_k, \alpha_k)) = a + k \cdot (b - a)/g + \alpha_k$$
$$\Delta(c) = (s_k, \alpha_k) \Leftrightarrow \Delta'((s_i, 0)) + \alpha_k = c$$ (8)

**2.2.3 Representation with two DT-2DLVs.** Let $(S \times H, \vee, \wedge, \rightarrow, ')$ be a two-dimensional linguistic lattice implication algebra for design teaching (DT-2DL-LIA) which is used to obtain two two-tuples of 2DLVs (DT-2DLVs), equivalent to a binary numerical array $(\beta_S, \beta_H)$. The adaptation of the design practice dimension is denoted as $\beta_S$ and the adaptation of the locality dimension is denoted as $\beta_H$. $(S \times H, \vee, \wedge, \rightarrow, ', (s_0, h_0), (s_g, h_t))$ is denoted as the two-dimensional linguistic lattice implication algebra for design teaching (DT-2DL-LIA), and the Hasse diagram is defined in this study, as shown in Fig 4.

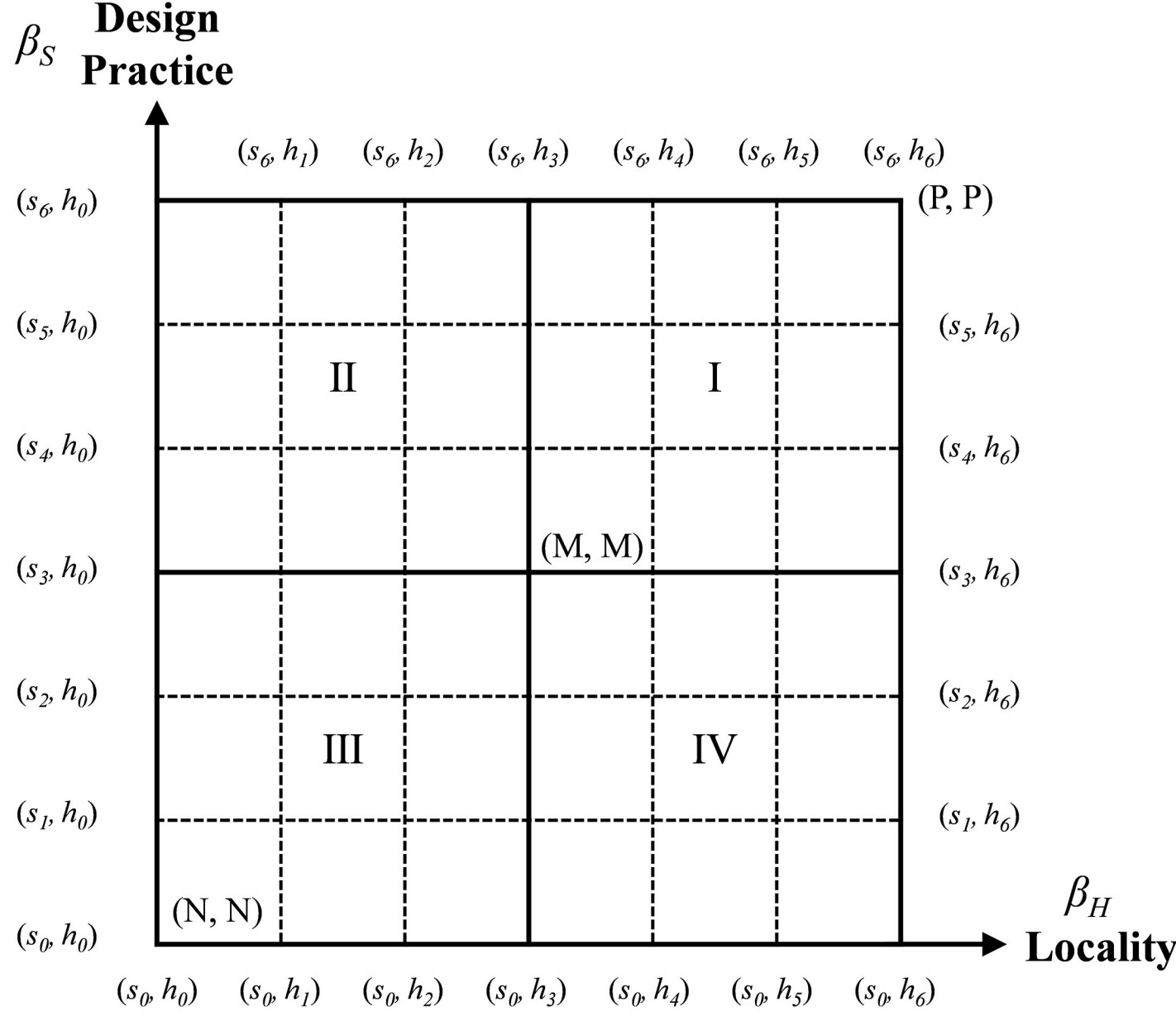

**Fig 4. Hasse Diagram of DT-2DL-LIA.**

DT-2DL-LIA is defined as:

$$\Delta : [0,g] \times [0,t] \to (S \times [-0.083, 0.083), H \times [-0.083, 0.083))$$
$$(\beta_S, \beta_H) \to \Delta(\beta_S, \beta_H)$$

(9)

Where $\Delta(\beta_S, \beta_H) = \Delta(\Delta_1(\beta_S), \Delta_2(\beta_H)) = ((s_{i1}, \alpha_{i1}), (h_{i2}, \alpha_{i2})$, The two functions are defined as $\Delta_1$ and $\Delta_2$. An inverse function $\Delta^{-1}$ of $\Delta(\beta_S, \beta_H)$ exists which maps a 2DLL with two two-tuples to

its equivalent binary array of values $(\beta_S, \beta_H) \in [0, g] \times [0, t]$, which is defined as follows:

$$
\begin{aligned}
&\Delta^{-1} : (S \times [-0.083, 0.083), H \times [-0.083, 0.083)) \rightarrow [0, g] \times [0, t]\\
&\Delta^{-1}((s_{i1}, \alpha_{i1}), (h_{i2}, \alpha_{i2}))\\
&= \left(\Delta_1^{-1}(s_{i1}, \alpha_{i1}), (h_{i2}, \alpha_{i2})\right)\\
&= (i1 + \alpha_{i1}, i2 + \alpha_{i2})\\
&= (\beta_S, \beta_H)
\end{aligned}
\tag{10}
$$

The original 2DLL can be represented as two two-tuples by adding 0 to each linguistic tag. Any semantic variable si can be transformed into a semantic binary of the form $(s_i, 0)$, and $h_i$ can be transformed into $(h_i, 0)$, i.e., $(s_i, h_j) = ((s_i, 0), (h_j, 0))$.

**2.2.4 Comparison method of two DT-2DLVs.** In the multi-attribute decision-making (MADM) process, the total order relationship between the two language labels needs to be defined. Therefore, to express the comparability and incomparability between two language labels, the relationship between DT-2DLVs is discussed below. The method for comparing the two DT-2DLVs is as follows. $(s_{i1}, h_{i2})$ and $(s_{j1}, h_{j2})$ are DT-2DL-LIAs, $i1, j1 \in [0-1]$ and $i2, j2 \in [0-1]$ are any two 2DLVs in the DT-2DL-LIA. The decision-maker's attitude towards assessing DT-2DLVs can be reflected by a positive number $\delta$. The parameter $\delta$ reflects the decision-maker's preference for self-assessment, and usually, the value of the parameter $\delta$ is less than 1, which can be predefined by the decision-maker at the beginning. If the decision-maker considers the assessment of the design practice dimension $(\beta_S)$ to be critical, then a smaller parameter $\delta$ can be set in the assessment, otherwise, a larger parameter $\delta$ can be set.

Using this approach, we can describe the comparable and non-comparable relationships between the alternatives.

1. If $i1 < j1$ and $i2 \leq j2$, or $i1 \leq j1$ and $i2 < j2$, then $(s_{i1}, h_{i2}) \leq (s_{j1}, h_{j2})$;

2. If $i2 \leq j2$ and $i1 - j1 < \delta$, then $(s_{i1}, h_{i2}) \leq (s_{j1}, h_{j2})$;

3. If $i1 = j1$ and $i2 = j2$, then $(s_{i1}, h_{i2}) = (s_{j1}, h_{j2})$;

4. If $i2 \leq j2$ and $i1 - j1 \geq \delta$, then $(s_{i1}, h_{i2})$ is incomparable to $(s_{j1}, h_{j2})$, denoted by $(s_{i1}, h_{i2}) \parallel (s_{j1}, h_{j2})$.

In addition, comparable relationships include both direct and indirect comparison, e.g.,

$\hat{r}_1 = (s_4, -0.002), (h_5, -0.011) = s_{0.668}, h_{0.819}$

$\hat{r}_2 = (s_4, 0.005), (h_5, -0.022) = s_{0.675}, h_{0.808}$

$\hat{r}_3 = (s_4, 0), (h_5, -0.029) = s_{0.670}, h_{0.801}$

$\hat{r}_4 = (s_4, 0.02), (h_5, -0.04) = s_{0.690}, h_{0.790}$

Set $\delta = 0.016$ and rank the alternatives. It can be known that when $\hat{r}1$ is compared with $\hat{r}_4$ and $\hat{r}_3$ with $\hat{r}_4$, $\delta > 0.016$ means that the two are not comparable, i.e., $\hat{r}_1 \parallel \hat{r}_4, \hat{r}_3 \parallel \hat{r}_4$. When $\delta \leq 0.016$, it means that the solutions of the relationship can be compared directly, i.e., $\hat{r}_1 > \hat{r}_2 > \hat{r}_3, \hat{r}_2 > \hat{r}_4$. Therefore, the indirect comparable relationship is $\hat{r}_1 > \hat{r}_4$, from which it can be inferred that $\hat{r}_1 > \hat{r}_2 > \hat{r}_4$. As shown in Fig 5, the dashed line indicates that there is no direct comparable relationship between the two solutions, and the symbol $\rightarrow$ indicates that there is a comparable relationship between them, and $\hat{r}_1 > \hat{r}_2 > \hat{r}_4$. and $\hat{r}_1 \rightarrow \hat{r}_2 = \hat{r}_1 > \hat{r}_2$.

**2.2.5 A two-dimensional linguistic weighted arithmetic aggregation (2DLWAA) operator with two two-tuples.** Dealing with ambiguous linguistic variables and aggregating expert group opinions usually requires aggregating their information, i.e., variable values. A common

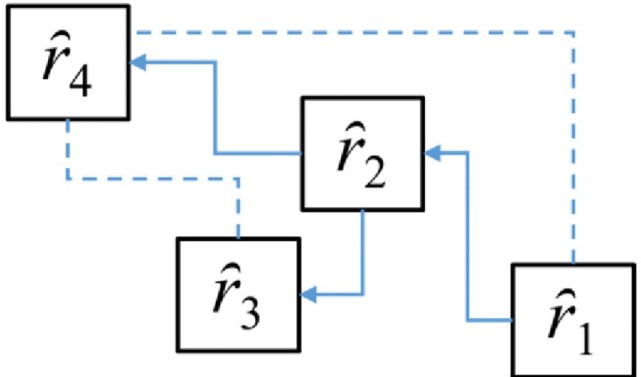

**Fig 5. Comparative relationship of two 2DLVs.**

approach is to convert these linguistic values into numbers, then aggregate them, and finally convert the numerical aggregation into semantic variables. In this study, a two-dimensional language-weighted operator aggregation (2DLWAA) operator with two binary groups is used [56], and then a comparison between two linguistic labels is defined in a DT-2DL-LIA based on the characteristics of the lattice. Then the 2DLWAA operator yields two integrated DLVs with two-tuples, both DT-2DLVs, as follows:

In this study, the applicability of the designed teaching method was defined as $\hat{r} = (s_{i1}, h_{i2})$, and the applicability assessment was performed by teachers in a team of experts, each of whom had their own opinion (positive, neutral, or negative) about the applicability of the teaching method. With n experts $E = \{e_1, e_2, ..., e_n\}$, $R = \{\hat{r}_1, \hat{r}_2, ..., \hat{r}_n\}$ is the set of DT-2DLVs (Fig 5) containing two two-tuples $(S \times H, \vee, \wedge, \rightarrow,)$, where $\hat{r}_i = (s_{i1}, h_{i2}) = ((s_{i1}, \alpha_{i1}), (h_{i2}, \alpha_{i2}))$, $s_{i1} \in S$, $h_{i2} \in H$, $\alpha_{i1}, \alpha_{i2} \in [-0.083, 0.083]$; $w_i = \{w_1, w_2..., w_n\}$ be the weight vector of $\hat{r}i$ such that $w_i \geq 0$ and $\sum_{i=1}^{n} w_i = 1$. For example, four experts $E = \{e_1, e_2, e_3, e_4\}$ are evaluated for suitability using $\hat{r}$ for design teaching method $D = \{d_1, d_2, d_3\}$ and the original evaluation value transforms the defuzzification number, and then obtains a two-dimensional linguistic decision matrix $R_{app} = (r_{ij})$ 4×3, representing the expert's perceived assessment of the pedagogical applicability of each method as follows:

$$R = \begin{pmatrix} & d_1 & d_2 & d_3 \\ e_1 & (s_{11}, h_{11}) & (s_{12}, h_{12}) & (s_{13}, h_{13}) \\ e_2 & (s_{21}, h_{21}) & (s_{22}, h_{22}) & (s_{23}, h_{23}) \\ e_3 & (s_{31}, h_{31}) & (s_{32}, h_{32}) & (s_{33}, h_{33}) \\ e_4 & (s_{41}, h_{41}) & (s_{42}, h_{42}) & (s_{44}, h_{44}) \end{pmatrix}$$

Then, 2DLWAA is called the two-dimensional linguistic weighted arithmetic aggregation (2DLWAA) operator.

$$R_{app} = 2DLWAA(\hat{r}_1, \hat{r}_2, ..., \hat{r}_n) = \Delta \left( \sum_{i=1}^{n} w_i \Delta^{-1}(\hat{r}_{ij}) \right)$$

$$= \Delta \left( \sum_{i=1}^{n} w_i \Delta_1^{-1}(s_{i1}, \alpha_{i1}), \sum_{i=1}^{n} w_i \Delta_2^{-1}(h_{i2}, \alpha_{i2}) \right) \qquad (11)$$

$$= \Delta \left( \sum_{i=1}^{n} w_i(i1, \alpha_{i1}), \sum_{i=1}^{n} w_i(i2, \alpha_{i2}) \right)$$

In [Eq 11], where $W = [w_i| \, i = 1,..., m]$ is a set of weight vectors, $w_i \in [0,1]$, $\sum w_i = 1$, and the functions $\Delta$ and $\Delta'$ are fuzzy linguistic-valued conversion functions.

## 3 Experimental and model evaluation results

### 3.1 Summary of the design thinking method

In this study, expert knowledge was utilized to assess the applicability of various design thinking methods in university education, and there were two stages in the process. In the first stage, to select a method that matches the curriculum of "locality" thematic design, a group of five teachers was organized to form a panel of experts. First, the leader of the focus group made a preliminary summary of more than 100 design thinking methods and organized them into a visual chart according to the six phases of design teaching ([Fig 6]), and then the leader of the group explained to the other members the characteristics of the methods as well as analyzed the process of their application, and then asked each member to express their understanding and opinion about each design thinking method, and finally all members discussed the advantages and disadvantages of each method and selected 71 methods suitable for the design course, which were summarized into 22 categories.

However, due to the similarity of types, purposes, and operational processes among the screened methods, only the objects targeted and the direction of development differed, such as

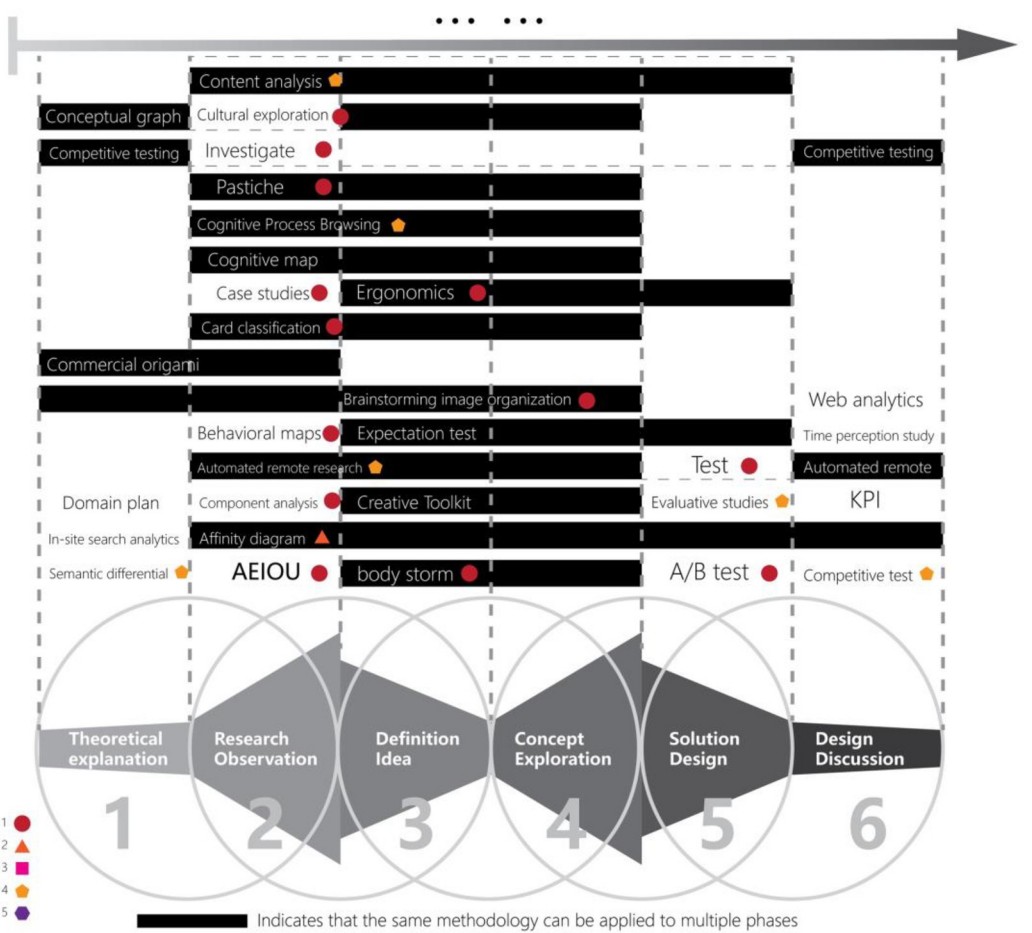

**Fig 6. Visualization charts for method selection (a small part).**

the observation and participatory observation methods, as well as the covert observation, shadow shape, and behaviour map. These five methods are all methods of observing environments, behaviours, and phenomena and collecting graphic data, but each has operational and application differences that are adapted to actual needs. For example, participant observation methods place more emphasis on the researcher's participation, while covert observation, in contrast, highlights the researcher's non-intervention, so they can be classified as types of observation methods. The panel of experts sub-categorized and tabulated the 71 methods by their characteristics (S1 Fig) and agreed that 22 types of methods have utility in teaching territorial design, and although some of them have rarely been utilized in the past, their conceptual sophistication and usability are considered to have potential applicability advantages.

## 3.2 Evaluation value of the teaching suitability of the design thinking

**3.2.1 The two-dimensional parameter $\delta$.** The parameter $\delta$ is an important value for ranking alternatives. This parameter can be predefined according to the decision-maker's preferences and the real uncertain environment. In this study, the value of $\delta$ was set according to the weighted evaluation of the dimensions of "design pedagogical practice" and "geographical theme" by the experts. The members of the group are $X = \{x_1, x_2, ..., x_n\}$ ($n = 15$), and the experts expressed their judgements independently based on their professional experience. The two-dimensional weighting questionnaire is used with a 100-point scale. The data results are shown in Fig 7, where the assessment data for 15 teachers were integrated and were standardized to design the teaching practice dimension $w_{DP} = 0.484$ and the locality theme dimension $w_L = 0.516$. $\delta = 0.5 - w_{DP} = 0.5 - 0.484 = 0.016$.

**3.2.2 Results of the evaluation of the applicability of teaching.** In the second stage, to analyze the applicability of the selected 22 categories of methods in the dimensions of "design practice" and "locality", an evaluation team consisting of 36 experts was formed to evaluate the methods by the Delphi method, and the specific steps were as follows:

1. The team leader described and explained the characteristics and application process of the 22 methods to the members.

2. The opinions of the members were collected using questionnaire statistics, and each member could view the evaluation opinions of other members and encourage mutual discussion.

3. The questionnaires were all returned and valid, and finally, the assessment values of each member were integrated and analyzed.

This stage of the assessment required the expert panel to enhance the understanding of the instructional methods through discussion to remove confusion and ambiguity. The suitability

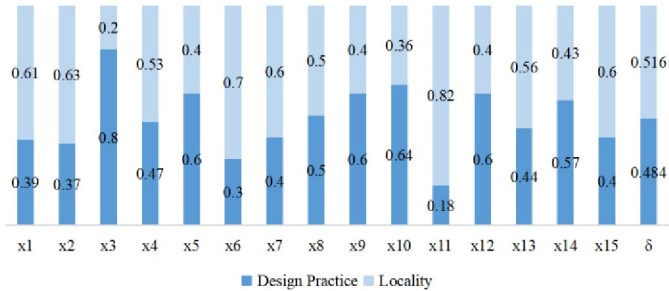

**Fig 7. δ-value of the dimensions of "design practice" and "locality".**

**Table 1. Results of the applicability evaluation of the 2-dimension linguistic fuzzy variables with 2-tuples of the 22 categories of design thinking.**

| NO. | $R_{DP}$ | $R_L$ | $\Delta^{-1}(\beta_S, \beta_H)$ | $\Delta(\beta_S, \beta_H)$ |
|---|---|---|---|---|
| $d_1$ | 0.669 | 0.706 | $s_{0.669}, h_{0.706}$ | $(s_4, -0.01), (h_4, 0.036)$ |
| $d_2$ | 0.704 | 0.676 | $s_{0.704}, h_{0.676}$ | $(s_4, 0.034), (h_4, 0.006)$ |
| $d_3$ | 0.670 | 0.801 | $s_{0.670}, h_{0.801}$ | $(s_4, 0), (h_5, -0.029)$ |
| $d_4$ | 0.681 | 0.743 | $s_{0.681}, h_{0.743}$ | $(s_4, 0.011), (h_4, 0.073)$ |
| $d_5$ | 0.657 | 0.832 | $s_{0.657}, h_{0.832}$ | $(s_4, -0.013), (h_5, 0.002)$ |
| $d_6$ | 0.668 | 0.819 | $s_{0.668}, h_{0.819}$ | $(s_4, -0.002), (h_5, -0.011)$ |
| $d_7$ | 0.691 | 0.703 | $s_{0.691}, h_{0.703}$ | $(s_4, 0.021), (h_4, 0.033)$ |
| $d_8$ | 0.690 | 0.790 | $s_{0.690}, h_{0.790}$ | $(s_4, 0.02), (h_5, -0.04)$ |
| $d_9$ | 0.662 | 0.793 | $s_{0.662}, h_{0.793}$ | $(s_4, -0.008), (h_5, -0.037)$ |
| $d_{10}$ | 0.678 | 0.781 | $s_{0.678}, h_{0.781}$ | $(s_4, 0.008), (h_5, -0.049)$ |
| $d_{11}$ | 0.686 | 0.753 | $s_{0.686}, h_{0.753}$ | $(s_4, 0.016), (h_5, -0.077)$ |
| $d_{12}$ | 0.675 | 0.753 | $s_{0.675}, h_{0.753}$ | $(s_4, 0.005), (h_5, -0.077)$ |
| $d_{13}$ | 0.666 | 0.798 | $s_{0.666}, h_{0.798}$ | $(s_4, -0.004), (h_5, -0.032)$ |
| $d_{14}$ | 0.691 | 0.729 | $s_{0.691}, h_{0.729}$ | $(s_4, 0.021), (h_4, 0.059)$ |
| $d_{15}$ | 0.676 | 0.780 | $s_{0.676}, h_{0.780}$ | $(s_4, 0.006), (h_5, -0.05)$ |
| $d_{16}$ | 0.702 | 0.689 | $s_{0.702}, h_{0.689}$ | $(s_4, 0.032), (h_4, 0.019)$ |
| $d_{17}$ | 0.675 | 0.808 | $s_{0.675}, h_{0.808}$ | $(s_4, 0.005), (h_5, -0.022)$ |
| $d_{18}$ | 0.683 | 0.768 | $s_{0.683}, h_{0.768}$ | $(s_4, 0.013), (h_5, -0.062)$ |
| $d_{19}$ | 0.692 | 0.723 | $s_{0.692}, h_{0.723}$ | $(s_4, 0.022), (h_4, 0.053)$ |
| $d_{20}$ | 0.714 | 0.636 | $s_{0.714}, h_{0.636}$ | $(s_4, 0.044), (h_4, -0.034)$ |
| $d_{21}$ | 0.701 | 0.724 | $s_{0.701}, h_{0.724}$ | $(s_4, 0.031), (h_4, 0.054)$ |

of the 22 categories of methods for designing instruction was assessed with expert panellists $X = \{x_1, x_2, . . ., x_n\}$ ($n = 36$). The applicability assessment questionnaire for the teaching methods was based on a seven-level semantic scale (Table 1) with numerical values indicating the degree of applicability. The results of the data for the assessment of the suitability of teaching methods based on the design of the dimensions of instructional practices are represented in Fig 8, which integrates the raw assessment values of the 36 teachers with a mean range of 4.889–6.194, with the highest values for the methods of $d_5$ (divergent thinking) and $d_6$ (information analysis), followed by $d_{17}$ (interview).

**3.2.3 Evaluation values of teaching applicability of two-dimensional linguistic fuzzy variables with two-tuples.** After converting the original assessment value of the teaching method applicability assessment into triangular fuzzy numbers, the assessment value of each

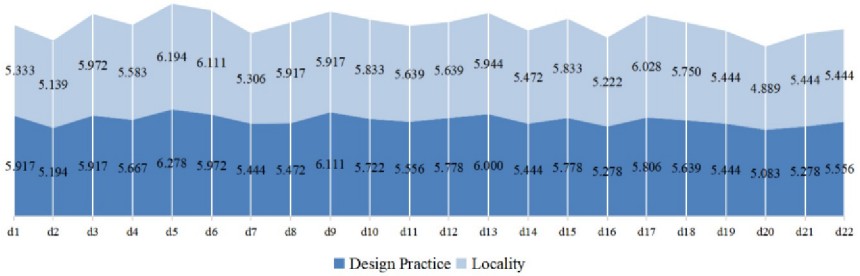

**Fig 8. The results for the assessment of the suitability of teaching methods.**

expert is first converted into triangular fuzzy numbers, and then the fuzzy numbers are approximated by the Euclidean distance method (Eq 2), and then the triangular fuzzy numbers can be subjected to the defuzzification operation by using Eq 3. The weight value of n-bit experts $w = 1/n = 1/36 \approx 0.028$ is set and data integration is performed with the 2DLWAA algorithm. As shown in Table 1, the defuzzification number of the applicability assessment value, both $\Delta^{-1}(\beta_S)$, for the $R_{DP}$ representation, considered from the perspective of designing the practice dimension of teaching, ranges in value from 0.657 to 0.714. The defuzzification number of the applicability assessment value, $\Delta^{-1}(\beta_H)$, for the $R_{DP}$ epresentation, considered from the perspective of the geographic thematic attributes, ranges in value from 0.636 to 0.832. $\Delta(\beta_S, \beta_H)$ denotes DLVs with two two-tuples (DT-2DL-LIA), and a function $\Delta^{-1}(\beta_S, \beta_H)$ exists that maps a 2DLL with two two-tuples to its equivalent array of binary values (DT-2DLVs).

The results of the design thinking approach to assessing pedagogical applicability can be expressed as a sequential relationship through the DT-2DL-LIA and are easy to understand intuitively. Fig 9a is divided into four regions: I, II, III, and IV. The DT-2DLVs of the 22

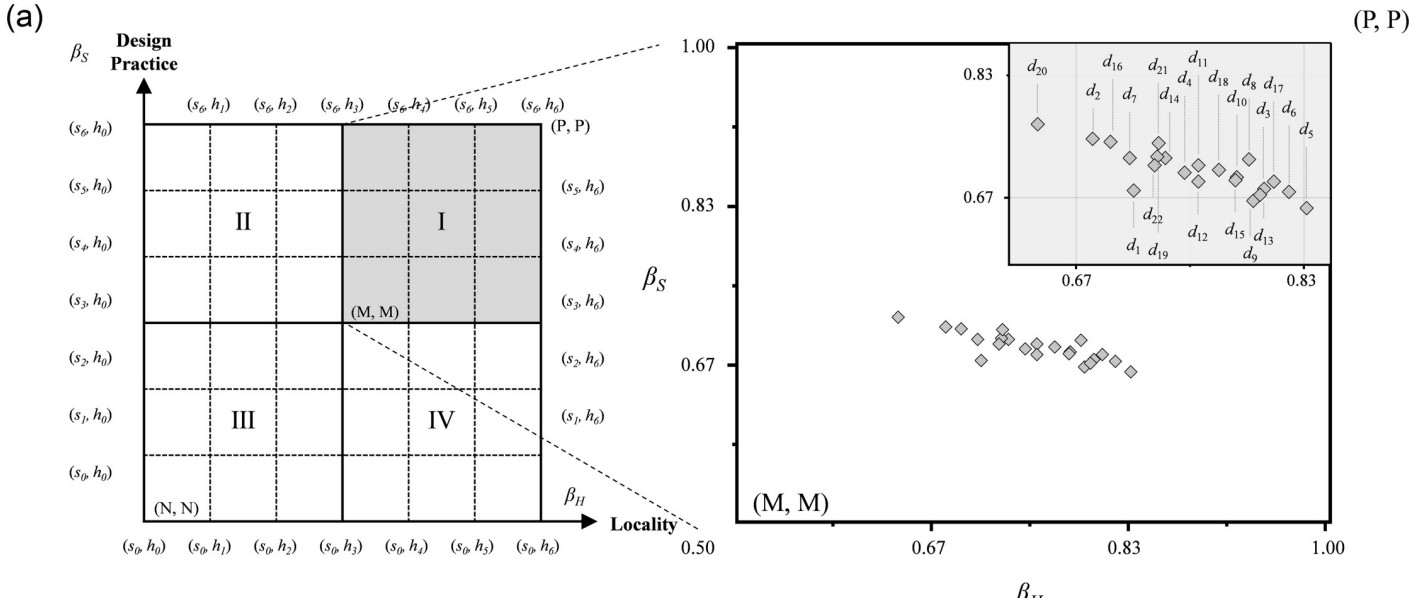

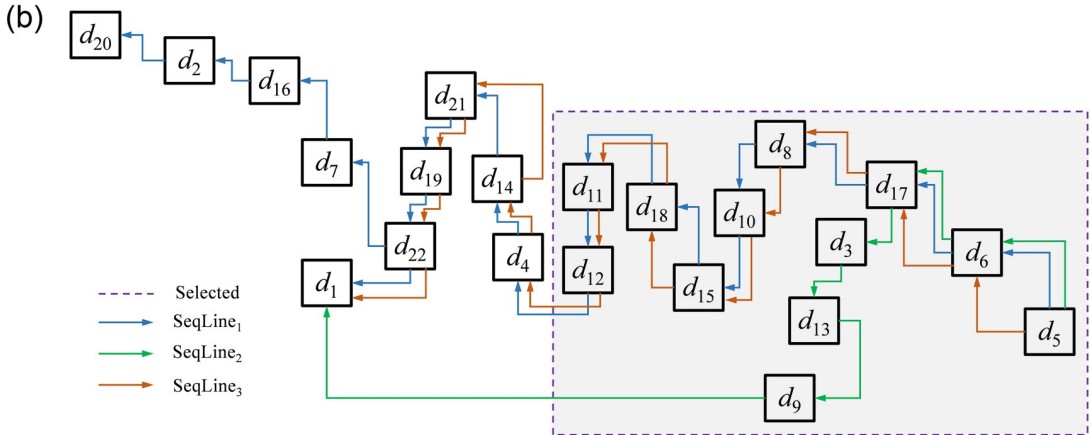

**Fig 9. DT-2DL-LIA sequential relationship diagram for the teaching applicability of design thinking.**

design thinking methods are all in region I of the lattice, which means that the applicability of all the methods to the teaching of design practice and locality is good. This means that all the methods have good applicability in teaching "design practice" and "locality". Fig 9b shows the order of applicability of all the design thinking methods in design teaching, and the topography is used to show the comparable high- and low-applicability relationships and the non-comparable relationships. As can be seen from Fig 9b, because some methods cannot be compared, two turning points (d17 and d22) are created, so three comparable sequential lines (including direct and indirect comparison) are formed, such as SeqLine1 = $\{d_5>d_6>d_{17}>d_8>d_{10}>d_{15}>d_{18}>d_{11}>d_{12}>d_4>d_{14}>d_{21}>d_{19}>d_{22}>d_7>d_{16}>d_2>d_{20}\}$, SeqLine2 = $\{d_5>d_6>d_{17}>d_8>d_{10}>d_{15}>d_{18}>d_{11}>d_{12}>d_4>d_{14}>d_{21}>d_{19}>d_{22}>d_1\}$, and SeqLine3 = $\{d_5>d_6>d_{17}>d_3>d_{13}>d_9>d_1\}$. Since each type of design thinking has its specific characteristics, advantages, and context, it is not comparable. For example, $d_3$ (feature categories) and $d_8$ (humanities exploration) are different types of design thinking, and feature categories are inductive behaviours from the bottom up, which not only help to clarify thoughts and build consensus but also effectively collect observations. Feature categories are bottom-up summarizing behaviours that not only help to clarify ideas, but also effectively collect observations and opinions, and graphically analyse them to identify common and important design issues. However, humanities exploration is an exploratory approach to understanding that delves deeper into the experience of understanding the user's world and obtaining ideas and insights from the user's own experience and is an inspirational tool that can better express an understanding of life, environments, concepts, and interactive behaviours. It is therefore difficult to compare the two approaches in terms of their pedagogical applicability, but only in relation to the contexts in which they are utilized and the roles they play.

## 4 Research discussion and teaching applications

### 4.1 Selection of design thinking approaches in localized design courses

The actual assessment and decision-making problems are generally complex things, and the multi-criteria assessment, although it can synthesize multiple factors to analyze the problem, is still one-dimensional, and the mutual exclusion and influence between the criteria belongs to the continuous multi-dimensional structure. In addition, the ambiguity and uncertainty of human thinking, as well as the finite nature of individual knowledge, experience, and ability to solve the decision-making problems faced often do not have all the necessary knowledge and information if they rely only on individual experience and wisdom. Although groups of experts have certain common behavioral and professional characteristics at a particular time, such as cultural level and value orientation. However, the fact that these characteristics of expert groups are often abstract, the form of evaluation given to alternatives is uncertain linguistic information, and the difficulty for individuals to understand the full picture of the group and the sometimes one-sided understanding of the decision-making goals makes it difficult for individual decision-makers to accurately grasp the behavioral characteristics of the group. In this study, thematicity and practicability were two important dimensions in determining the appropriateness of the pedagogical methods used in the design course, and the two were not isolated, but rather in a state of interaction, which added to the complexity of the decision. When assessing the thematic dimension also affects the practicality, therefore, the method proposed in this study is a continuous linguistic representation of the evaluation information, which is more suitable for the 2 dimensions and avoids missing and distorted information. The preferred approach through visualization of graphs is more intuitive and precise than numerical. The main objective is to screen and develop appropriate design teaching methods

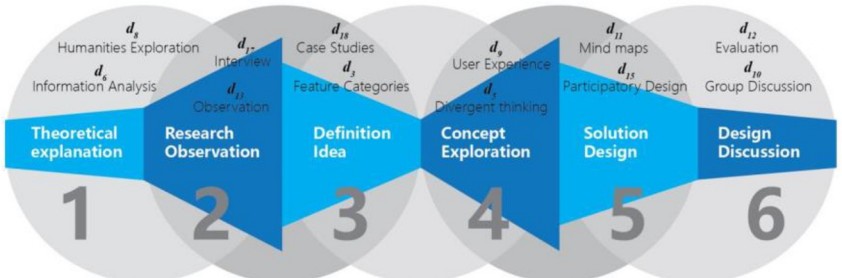

**Fig 10. 12 Types of design thinking methods corresponded and used in the 6 stages of design teaching.**

by collecting group opinions from university teachers with rich design teaching experience, analyzing and evaluating them, and identifying the knowledge of the expert group.

As each type of design thinking method has suitable application methods and features to target the various stages of design teaching courses, the DT-2DLVs will be utilized to assess their effectiveness. Based on the ordering of the high and low values of the DT-2DLVs and the characteristics of the geographically based design courses, 12 categories of design thinking methods with higher pedagogical applicability were selected, as shown in the dotted area in Fig 9b, and the results of the selections represent the professional judgement and group decision-making of each instructor of the expert group on the applicability of each pedagogical method. Fig 10 represents the application of the 12 categories of design thinking methods to the six phases of design instruction, with two categories of methods that could be utilized in each phase.

Based on the results of this research, the design thinking methods that were considered by the experts to be of high applicability were also those that were often used in design teaching, such as divergent thinking represented by brainstorming, and observational research methods represented by field research and observation. Such results were produced mainly since experts are easily influenced by their own teaching experience, so if new attempts are desired, more attention needs to be paid to methods that have a slightly lower assessment value among the methods screened, such as humanistic inquiry and participatory design, as well as the case study methodology and the interview and survey method with assessment. These methods may have some advantages that have been overlooked by experts due to their lesser use and therefore have some potential that can be explored, developed, and applied.

## 4.2 Design thinking applied to a localized design course

To verify the applicability of the design thinking method selected and developed in this paper in design teaching, we have applied it in the curriculum of a university environmental design programme, using the environmental renovation design course as a case study to transform the design thinking method into a teaching method. The course will continue to focus on the integration of traditional design into new residential and commercial spaces, as well as designs for traditional elements and regional cultural displays and landscapes, as well as the architectural renovation of traditional historical neighbourhoods and ancient houses, and research and design for village improvement. This will involve the comprehensive use of classroom teaching, case study teaching, project teaching, etc., to play a guiding role for teachers and mobilize students' learning enthusiasm, initiative, and creativity. It will also encourage students to link theory to practice, and cultivate their ability to identify, analyse and solve

problems. They will interact with local communities through design practice, taking residents or users as the core subjects, and promote the sense of identity of locality development and place.

The following sections are based on the teaching characteristics of the curriculum and follow the six stages shown in Fig 10. Each stage is based on a cyclic development model, that is, when proceeding from one stage to the next, if it is found that the purpose cannot be reached, the previous stage can be repeated again, and when the goal is accomplished, the next one can be started. After six rounds of teaching and learning practice in 2018–2022, the teaching methods suitable for the university design course have been summarized, and the students have been able to produce certain positive results in their learning.

**4.2.1 Theory lecture.**   The first stage of the design for environmental renovation course is a theoretical presentation, which involves explaining the background, aims, and concepts of the course to students, and developing teaching and learning activities based on practical design projects. It is necessary to build up students' understanding of the content of the course, such as the concept of locality, the development of locality design, and related cases, so humanistic inquiry and information analysis are combined and applied (Fig 11). Firstly, the concept of locality is explained in terms of the geographical environment, material resources, historical development, folk culture, and lifestyle of the site; then the development of locality design is explained through case studies from different periods around the world. Finally, in-depth experience is gained from the user's perspective to obtain first-hand experiences, thoughts, and insights, which can be used to express an understanding of life, the environment, the concept, and interactive behaviour. This course will provide students with a certain knowledge base before they are ready to start learning and practising the content of the course and will require them to actively search for places and design cases with regional characteristics and to utilize information analysis methods to generalize and understand them from the perspectives mentioned above.

**4.2.2 Research observation.**   After students have gained an initial comprehension of the concept of locality, and the characteristics of venues and users, they only have a limited amount of past knowledge and experience, and many of their understandings are not yet deep enough. Therefore, in the second stage of the course, it is necessary to develop their learning with clear target venues (to be drawn up by the teacher or the students) to move from theoretical understanding to practical exploration. This involves careful observation of various

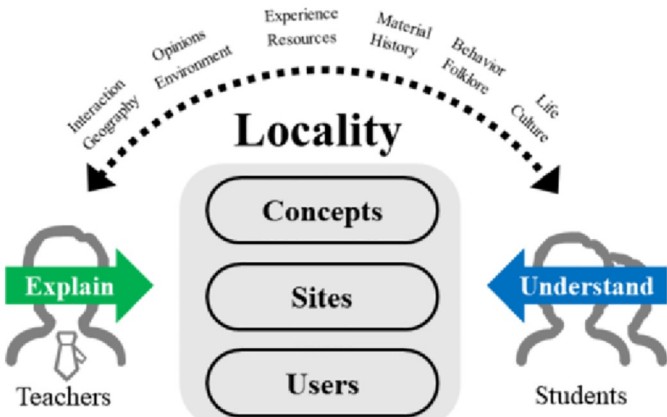

**Fig 11. Theoretical presentation mode of design courses.**

phenomena in the venue and making systematic recordings (graphic and audio), including observing people, objects, environments, events, behaviours, and interactive processes, but also understanding situations and people's behaviours by engaging in activities, situations, cultures, and subcultures (Fig 12a). In addition, interviews with users are tools for gathering descriptive information material in written form to understand the characteristics, thoughts, behaviours, perceptions, opinions, or attitudes of the respondents about a topic. These are used along with in-depth site observations to explore design problem points (Fig 12b).

**4.2.3 Definition of ideas.** After students have gained a deeper understanding of the venue and users, they need to further analyse the problem points and explore the ideas for design development based on them. Therefore, in the third stage of the course, it is necessary to define the problem points and the direction of design development. Due to the students' lack of professional experience, they are not yet able to effectively transform the problem points obtained from observations and interviews into design ideas. The analysis of relevant design cases can provide students with some inspiration and summarize the design background, purpose, concept, venue, application of elements, style, function, structure, material, user, and other

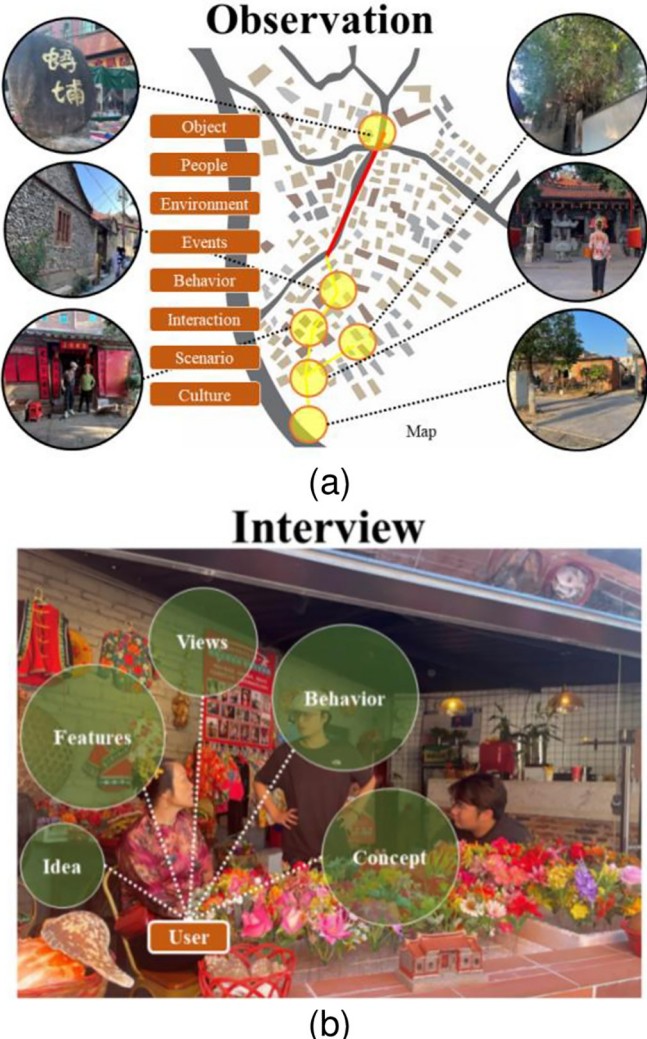

(a)

(b)

**Fig 12. Site observations and user interviews.** (Street view source: photographs taken by the paper's researchers).

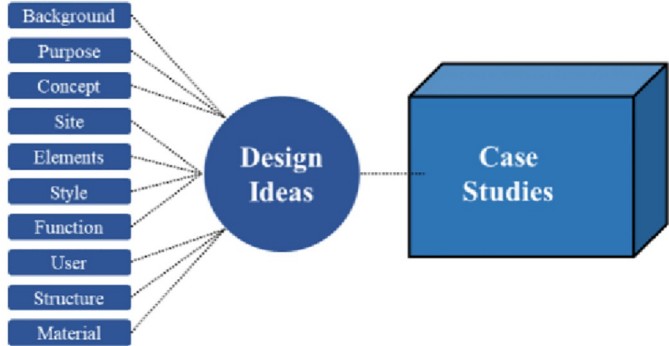

**Fig 13. Case studies and characterization methods.**

characteristics of the cases, which can be used to analyse the key points of the design and realize the establishment of design ideas concerning past experiences (Fig 13).

**4.2.4 Concept exploration.** When the basic design ideas are identified, it is necessary to gradually expand these into the possible development of design concepts. This stage will be conducted with the help of divergent thinking using user empathy and experience mapping methodology, which captures the user's listening, speaking, doing, and thinking, as well as their needs and feelings when accomplishing a certain task or realizing a certain goal utilizing an empathetic pathway to synthesize and analyse the touchpoints, contexts, users, behaviours, emotions, and target flow. They will develop core concepts and identify features, arguments, and related ideas in a collaborative team approach (design team) for a specific topic, identifying the core pain points and design opportunities (Fig 14).

**4.2.5 Solution design.** As a result of the previous phase of analysis, each student design team will have a clear point of opportunity to use as inspiration to move into the schematic design phase. The students will work through specific designs of form, structure, function, materiality, interaction, and flow to solve problems and reach their goals (Fig 15). To facilitate discussion, development, and the creation of new solutions, imagery mapping is employed as a visual way to stimulate participation, find patterns and concepts, and guide the positive evaluation of the subject matter to help designers consider the context of the use of technology and form factors from multiple perspectives. This process promotes the designer's creative insight and expression of their emotions, dreams, needs, and desires, providing a rich source of information for conceptual design. The students utilize two-dimensional/three-dimensional

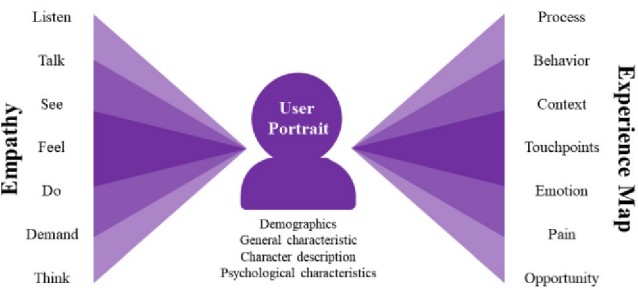

**Fig 14. User empathy and experience map.**

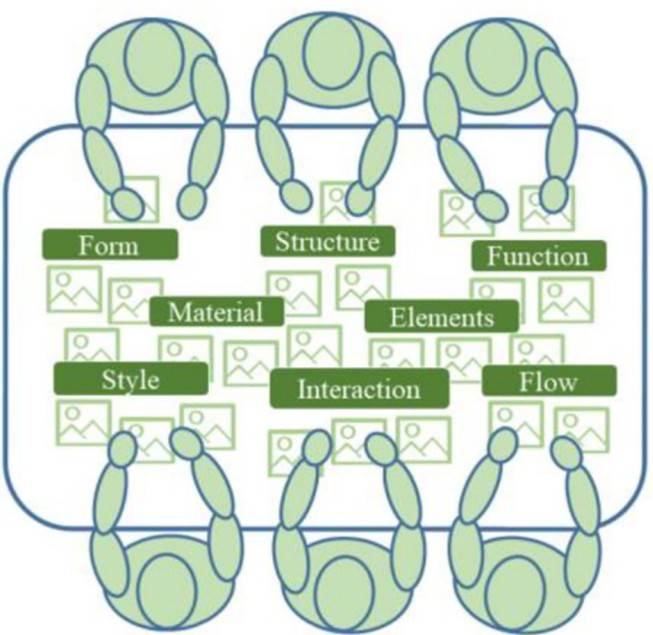

**Fig 15. Imagery and participatory design.**

drawings (including plans, sections, elevations, analyses, and renderings) or prototypical representations to propose relatively rational solutions.

**4.2.6 Design discussion.** Whether the design solutions proposed by the students achieve rationality, usefulness, validity, feasibility, and a satisfactory purpose of the assignment requires a check regarding the effectiveness of the design. The method of group discussion is similar to a mini-seminar, where students present creative design solutions through exhibitions. So that the programme can be developed in a better way, it relies on the dynamism created by the expert assessment team to discuss and evaluate the students' programmes and explore ideas for improvement. The assessment indicators were categorized into site and user research (SU), development of ideas (DI), divergent concepts (DC), problem-solving (PS), and creative presentation (CP), with 20 impact assessment points allocated for each one (Fig 16) and a total score of 100. If the final score for the evaluation was below 60, the design effectiveness of the programme could not be achieved. In addition, each expert also suggests modifications for each programme, and students can make step-by-step adjustments based on the comments.

## 5 Conclusions

This method for obtaining an objective assessment through group decision-making has been widely proven to be reliable, valid, and reasonable, and the use of fuzzy theory can refine the assessment values, reduce the loss of information, and express the expert's opinion more accurately. In this paper, we propose a two-dimensional linguistic fuzzy model with two-tuples based on group decision-making, taking the assessment of the applicability of design thinking in teaching as a case study, and then apply the design thinking method with higher applicability to the environmental remodelling design courses of university design majors. The validation through years of teaching practice shows that the computational model proposed is easy

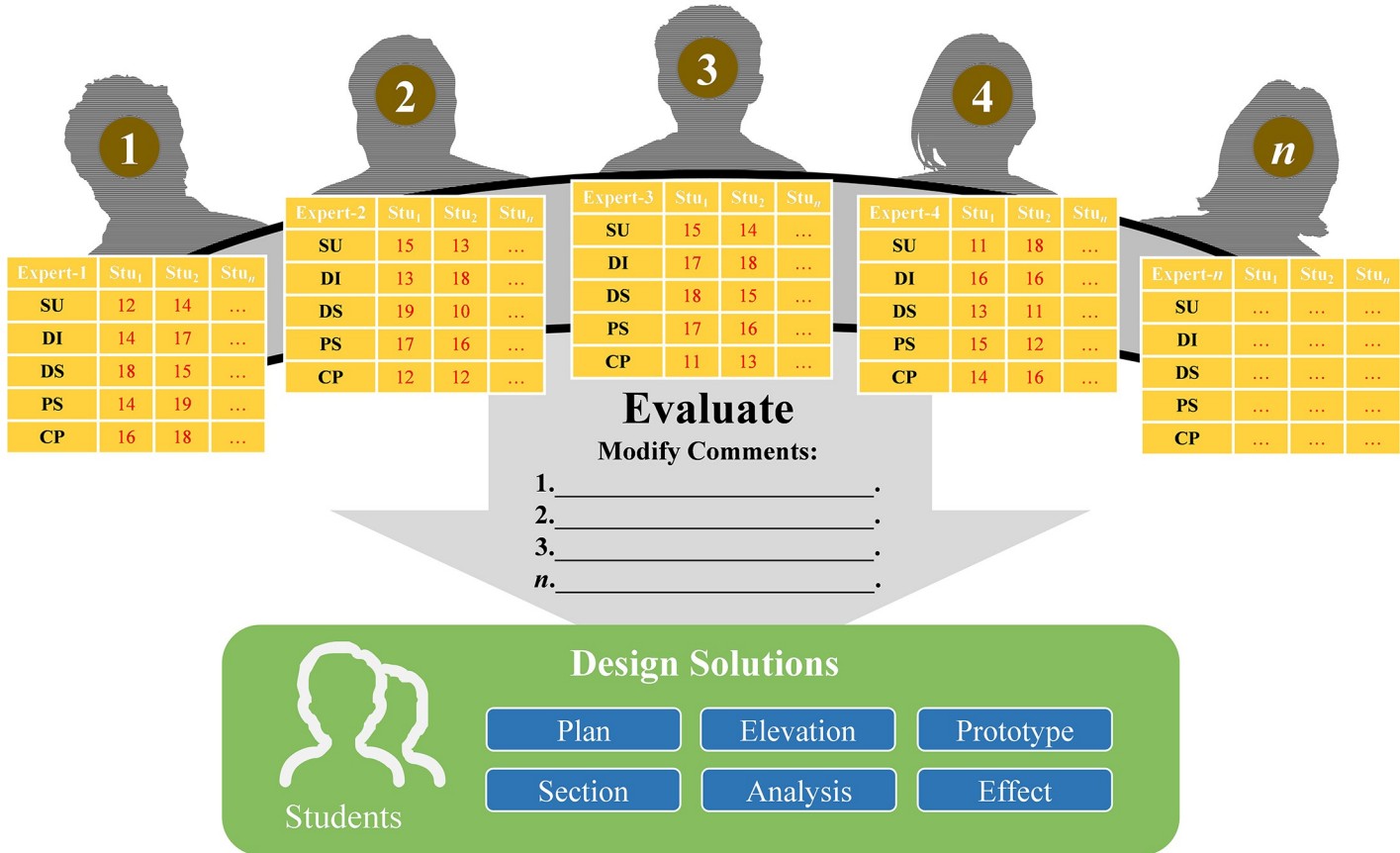

**Fig 16. Expert group discussions and assessments.**

to operate, improves the accuracy of assessment decisions, and has excellent operationalization, a wide range of adaptability, and clear semantics.

Through the literature review and actual teaching experience, it was found that design courses contain two-dimensional attributes. The first is the subject matter, which, in addition to the "locality" dimension described in this study, may involve other types, such as intelligence, entrepreneurship, and marketing, etc. The second is the pedagogical mode of the course, which may include participation in exploration, discussion, and experimentation as well as "design practice". Perceived reality representation is a qualitative aspect, which requires linguistic variables. We utilize two-tuple linguistic models to avoid the problem of the loss of information suffered by some fuzzy linguistic approaches when aggregating linguistic variables. Evaluations of each teaching method were conducted based on the aggregation of these perceptions, and these facilitated the decision-making process. The proposed assessment method requires experts to have enough experience in design teaching as well as a more comprehensive understanding of the types and application characteristics of design thinking methods, but there are some problems in the research process, such as the evaluators lacking application experience in some infrequently used methods. As a consequence, this may affect their judgement regarding the potential of the future application of such methods, leading to the results of the affected assessment, for example, in the cases of "stakeholder studies" and "data interpretation". Therefore, this assessment method is advantageous for the integration

and transformation of existing design thinking methods, but has limitations for the innovation and development of new ones.

In future research, we plan to extend the applicability of binary fuzzy semantic-based analysis of teaching methods in two directions. On the one hand, we intend to incorporate other decision-making heuristics into the studied system so that they can handle fuzzy linguistic information. Teaching methods and processes are almost dynamic in the actual teaching of design courses. For example, if the students do not learn well at a certain stage of the course, it is necessary to return to the previous step of the course or to change some of the methods to adapt them to the students' ability to master the knowledge. In addition, the relatively high level of dynamism may more realistically model an expert's decision-making strategy, as teaching methods need to be constantly added or adapted to accommodate a combination of online and offline modes of teaching and learning. On the other hand, each type of design thinking method has diverse usage characteristics that need to be continuously adapted and optimized in teaching practice. Therefore, we will continue to validate them through additional design teaching practices, which can be used to develop new teaching methods that are adaptable to a wide range of educational systems and have a high degree of innovation and practicality.

## Supporting information

**S1 Fig. 22 categories of design thinking methods.**
(DOCX)

## Author Contributions

**Conceptualization:** You-Lei Fu.

**Data curation:** Ruoqi Dai.

**Formal analysis:** Linxin Zheng, Ruoqi Dai.

**Funding acquisition:** Linxin Zheng.

**Investigation:** You-Lei Fu.

**Methodology:** You-Lei Fu.

**Project administration:** Kuei-Chia Liang.

**Resources:** Linxin Zheng.

**Supervision:** Kuei-Chia Liang.

**Visualization:** Ruoqi Dai.

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
