## [Decision Letter · Decision Letter 0]

26 Dec 2023

PONE-D-23-35366How to choose the design thinking method in teaching the design of localization: 2-dimension linguistic fuzzy model with 2-tuplesPLOS ONE

Dear Dr. Fu,

Thank you for submitting your manuscript to PLOS ONE. After careful consideration, we feel that it has merit but does not fully meet PLOS ONE’s publication criteria as it currently stands. Therefore, we invite you to submit a revised version of the manuscript that addresses the points raised during the review process.

Please follow comments of the reviewers to further improve the quality of your paper.

We look forward to receiving your revised manuscript.

Kind regards,

Ta-Chung Chu

Academic Editor

PLOS ONE

4. We note that Figures 11 and 12 in your submission contain copyrighted images. All PLOS content is published under the Creative Commons Attribution License (CC BY 4.0), which means that the manuscript, images, and Supporting Information files will be freely available online, and any third party is permitted to access, download, copy, distribute, and use these materials in any way, even commercially, with proper attribution. For more information, see our copyright guidelines: http://journals.plos.org/plosone/s/licenses-and-copyright.

 1. You may seek permission from the original copyright holder of Figures 11 and 12 to publish the content specifically under the CC BY 4.0 license.

5. We note that Figure 11 in your submission contain [map/satellite] images which may be copyrighted. All PLOS content is published under the Creative Commons Attribution License (CC BY 4.0), which means that the manuscript, images, and Supporting Information files will be freely available online, and any third party is permitted to access, download, copy, distribute, and use these materials in any way, even commercially, with proper attribution. For these reasons, we cannot publish previously copyrighted maps or satellite images created using proprietary data, such as Google software (Google Maps, Street View, and Earth). For more information, see our copyright guidelines: http://journals.plos.org/plosone/s/licenses-and-copyright.

1. You may seek permission from the original copyright holder of Figure 11 to publish the content specifically under the CC BY 4.0 license. 

Reviewers' comments:

Reviewer's Responses to Questions

**Comments to the Author**

1. Is the manuscript technically sound, and do the data support the conclusions?

Reviewer #1: Partly

Reviewer #2: Yes

2. Has the statistical analysis been performed appropriately and rigorously? 

Reviewer #1: N/A

Reviewer #2: Yes

3. Have the authors made all data underlying the findings in their manuscript fully available?

Reviewer #1: No

Reviewer #2: Yes

4. Is the manuscript presented in an intelligible fashion and written in standard English?

Reviewer #1: No

Reviewer #2: Yes

5. Review Comments to the Author

Reviewer #1: The article titled "How to choose the design thinking method in teaching the design of localization: 2-dimension linguistic fuzzy model with 2-tuples" embarks on an ambitious journey to bridge the gap between design thinking methodologies and their practical application in university-level design education. The core of this study revolves around the innovative application of a 2-dimension linguistic fuzzy model paired with 2-tuples. This approach aims to evaluate the suitability of various design thinking methods, taking into account critical factors such as design practice, locality, the educational philosophy of design, and the unique characteristics of the student body.

A notable strength of this research lies in its methodological rigor and the involvement of 36 experts in the field, which lends a degree of robustness to the findings. The use of established statistical methods, including Delphi, triangular fuzzy number, and Euclidean distance method, further reinforces the credibility of the study. This approach not only caters to the academic community but also holds practical value, as it seeks to directly influence teaching methodologies in design education.

However, the article is not without its shortcomings. A significant issue is the clarity of language. The complexity of the subject matter necessitates precise and clear communication, which could be greatly improved with the assistance of a native English speaker. This would enhance the readability and overall accessibility of the article.

Moreover, the study grapples with a lack of clear novelty. While the approach is undoubtedly interesting, the article struggles to differentiate its contributions from existing works in the field. A more explicit articulation of what sets this study's approach apart is crucial. This ties in with another weakness: the scope of the literature review. Currently, the review does not sufficiently contextualize the study within the broader landscape of design education and design thinking. An expanded literature review could provide a more comprehensive background, setting the stage for the study's contributions more effectively.

Additionally, the article falls short in clearly defining its novelty and potential impact on the field of design education. There is a need for a more articulate presentation of the advantages and limitations of the proposed model. This could be further bolstered by a comparative analysis with other existing models or methods in the teaching of design thinking, offering a richer, more nuanced understanding of the study's place within the field.

The broader implications of the study's findings also require further elaboration. Discussing how these findings could influence design education beyond the specific context of the study would add depth to the research and highlight its potential significance.

In conclusion, the study presents a valuable and interesting methodological approach to a pertinent issue in design education. However, to truly realize its potential and make a significant contribution to the field, the article requires major revisions. These include enhancing the clarity of language, expanding the literature review, articulating the study's unique contributions and potential impact more clearly, and providing a broader context for its implications. With these revisions, the study has the potential to become a noteworthy addition to the discourse on design thinking in education.

Reviewer #2: The paper is very interesting, and the idea is novel. It is a great idea. I have only one comment about the English in the paper. Please review it again, as there are some spelling and punctuation errors. I think it is suitable for publication after that.

6. PLOS authors have the option to publish the peer review history of their article (what does this mean?). If published, this will include your full peer review and any attached files.

Reviewer #1: No

Reviewer #2: No

---

## [Author Response · Author response to Decision Letter 0]

21 Jan 2024

Responds to the reviewer’s comments:

We are very grateful for the reviewers' comments and will respond to each question in the "Reviewer Questions and Responses" section.

Reviewer 1: 

(1)The problem of improving the clarity of language and grammar.

Response: We are very grateful to your comments for the manuscript. According to your advice, We have sent the article to a professional organization for English proofreading and have provided supporting documents.

(2)Recommendations for expanding the literature review.

Response: We appreciate and accept the reviewer's suggestions and have included "1.2 Locality and Design Teaching" and "1.2 Locality and Design Teaching" in the supplement some of the literature. In addition, "1.3 Selection of Teaching Methods" has been added to clarify the novelty of this study and to differentiate it from other studies in the field. For details, please refer to "1 Introduction".

Supplementary Literature:

Wu, Y. J., & Chen, J. C. (2021). Stimulating innovation with an innovative curriculum: a curriculum design for a course on new product development. The International Journal of Management Education, 19 (3), 100561.

Béchard, J. P., & Grégoire, D. (2005). Entrepreneurship education research revisited: The case of higher education. Academy of management learning & education, 4(1), 22-43.

Nabi, G., Liñán, F., Fayolle, A., Krueger, N., & Walmsley, A. (2017). The impact of entrepreneurship education in higher education: A systematic review and research agenda. Academy of management learning & education, 16(2), 277-299.

Manimala, M. J., & Thomas, P. (2017). Entrepreneurship education: innovations and best practices. Entrepreneurship education: Experiments with curriculum, pedagogy and target groups, 3-53.

Rasiah, R., Somasundram, S., & Tee, K. P. (2019). Entrepreneurship in education: Innovations in higher education to promote experiential learning and develop future ready entrepreneurial graduates. development, 6, 7.

Tan, T. A. G., & Vicente, A. J. (2019). An innovative experiential and collaborative learning approach to an undergraduate marketing management course: A case of the Philippines. The International Journal of Management Education, 17(3), 100309.

Thijs, A., & Van Den Akker, J. (2009). Curriculum in development. Netherlands Institute for Curriculum Development (SLO).

Powers, L. M., & Summers, J. D. (2009). Integrating graduate design coaches in undergraduate design project teams. International Journal of Mechanical Engineering Education, 37(1), 3-20.

Bernstein, W. Z., Ramanujan, D., Zhao, F., Ramani, K., & Cox, M. F. (2012). Teaching design for environment through critique within a project-based product design course. International Journal of Engineering Education, 28(4), 799.

De Vere, I., Melles, G., & Kapoor, A. (2010). Product design engineering–a global education trend in multidisciplinary training for creative product design. European journal of engineering education, 35(1), 33-43.

Eggink, W. (2009). A practical approach to teaching abstract product design issues. Journal of engineering design, 20(5), 511-521.

Samuel, A. B., & Rahman, M. M. (2018). Innovative teaching methods and entrepreneurship education: A review of literature. Journal of Research in Business, Economics and Management, 10(1), 1807-1813.

Goodyear, P. (2015). Teaching as design. Herdsa review of higher education, 2(2), 27-50.

Marjanovic, O. (2016). Designing innovative education through action design research: Method and application for teaching design activities in large lecture environments. Journal of Information Technology Theory and Application (JITTA), 17(2), 1.

Cameron, L. (2017). How learning designs, teaching methods and activities differ by discipline in Australian universities. Journal of learning design, 10, 69-84.

(3)Clearer articulation of the unique contributions and potential impacts of the research.

Response: Thanks to the reviewers' approval and comments on this study, we found by analyzing the past studies that the current methods of assessing the applicability of teaching methods from the literature analysis to the statistical analysis are the results of one dimension, and do not take the thematic characteristics of the course as the second dimension. and therefore the assessment of suitability is not comprehensive enough. For this reason, this study proposes a 2-dimensional expert assessment method, setting up two concepts of thematic character of the course and pedagogical mode, and adopting a 2-dimension linguistic fuzzy model with 2-tuples to establish mixed statistical methods, including Delphi method, triangular fuzzy number method, Euclidean distance method, and 2DLL method, 2DLWAA method.

We add the unique contributions and potential impacts of the research described above to "1.3 Selection of Teaching Methods" in the article.

(4)Provide a broader context for the impact of research.

Response: Through the literature review and actual teaching experience, it was found that design courses contain 2-dimensional attributes: first, the subject matter of the course, in addition to the "locality" subject matter described in this study, there may be other types of subject matter, such as intelligence, entrepreneurship, and marketing, etc. Second, the teaching mode of the course, in addition to "design practice", there may be participatory exploration, discussion, and experimentation. The second is the pedagogical mode of the course, which may include participation in exploration, discussion, and experimentation in addition to "design practice". In addition, the linguistic fuzzy model with 2-tuples is used to find the appropriate design thinking method, which makes the assessment process more comprehensive and avoids the problem of information loss. The assessment method proposed in this study requires experts to have enough experience in design teaching as well as a more comprehensive understanding of the types and application characteristics of design thinking methods, but there are some problems in the research process, such as the lack of application experience of certain experts in some infrequently used methods, and therefore the lack of sufficient judgment of the potential of the future application of such methods, which leads to the results of the affected assessment, for example, "Stakeholder studies" and "Data Interpretation". Therefore, this assessment method is advantageous for the integration and transformation of existing design thinking methods, while it has limitations for the innovation and development of thinking methods.

We add the implications of the research described above in "5 Conclusions".

Reviewer 2: 

1.Some spelling and punctuation error issues in the article. 

Response: We seriously thank the reviewers for their work and comments. According to your advice, We have sent the article to a professional organization for English proofreading and have provided supporting documents.

---

## [Decision Letter · Decision Letter 1]

15 Feb 2024

PONE-D-23-35366R1How to choose a design thinking method for teaching the design of localization: A two-dimension linguistic fuzzy model with two-tuplesPLOS ONE

Dear Dr. Fu,

Thank you for submitting your manuscript to PLOS ONE. After careful consideration, we feel that it has merit but does not fully meet PLOS ONE’s publication criteria as it currently stands. Therefore, we invite you to submit a revised version of the manuscript that addresses the points raised during the review process.

Please respond to the comments provided by Reviewer #1 to further improve your manuscript.

We look forward to receiving your revised manuscript.

Kind regards,

Ta-Chung Chu

Academic Editor

PLOS ONE

Journal Requirements:

Reviewers' comments:

Reviewer's Responses to Questions

**Comments to the Author**

1. If the authors have adequately addressed your comments raised in a previous round of review and you feel that this manuscript is now acceptable for publication, you may indicate that here to bypass the “Comments to the Author” section, enter your conflict of interest statement in the “Confidential to Editor” section, and submit your "Accept" recommendation.

Reviewer #1: (No Response)

Reviewer #2: All comments have been addressed

2. Is the manuscript technically sound, and do the data support the conclusions?

Reviewer #1: Yes

Reviewer #2: Yes

3. Has the statistical analysis been performed appropriately and rigorously? 

Reviewer #1: N/A

Reviewer #2: Yes

4. Have the authors made all data underlying the findings in their manuscript fully available?

Reviewer #1: No

Reviewer #2: Yes

5. Is the manuscript presented in an intelligible fashion and written in standard English?

Reviewer #1: Yes

Reviewer #2: Yes

6. Review Comments to the Author

Reviewer #1: In the paper "Analysis of the Applicability of Design Thinking Methods in Design Education at Universities," the authors attempt to assess the applicability of various design thinking methods in the context of university education. The authors have clearly made efforts to improve the article, but there are still ambiguities regarding how expert knowledge was identified. It has not been sufficiently explained how exactly this knowledge was identified, which can be problematic for readers trying to understand the research process.

At the same time, it is worth considering whether uncertainty could have been modeled in a different way, perhaps more suitable for the analyzed problem. The authors should consider whether there are other methods of identifying expert knowledge that could have been used in this context, and whether the methodology employed, such as the RANCOM method, could have been more appropriate or simply more accessible to readers.

Furthermore, there is potential to increase attention to the sensitivity associated with the applied uncertainty generalization. The authors should consider the possibility of more detailed discussion of this issue to provide readers with a fuller understanding of the analyzed results.

After addressing the above comments, the article will be ready for acceptance for publication.

Reviewer #2: Authors did all corrections in good manor. I recommend accepting the paper in revised form. Best wishes.

7. PLOS authors have the option to publish the peer review history of their article (what does this mean?). If published, this will include your full peer review and any attached files.

Reviewer #1: No

Reviewer #2: No

---

## [Author Response · Author response to Decision Letter 1]

24 Feb 2024

Dear Editors and Reviewers:

Thank you for your letter and for the reviewers’ comments concerning our manuscript entitled “How to choose a design thinking method for teaching the design of localization: A two-dimension linguistic fuzzy model with two-tuples” (ID: PONE-D-23-35366R1). Those comments are all valuable and very helpful for revising and improving our paper, as well as the important guiding significance to our researches.

We have carefully studied these comments and made minor revisions which we hope will be approved. Revised portion are marked in red in the paper. The main corrections in the paper and the responds to the reviewer’s comments are as flowing.

Responds to the reviewer’s comments:

Reviewer 1: 

1.The authors have clearly made efforts to improve the article, but there are still ambiguities regarding how expert knowledge was identified. It has not been sufficiently explained how exactly this knowledge was identified, which can be problematic for readers trying to understand the research process.

Response: We accept the reviewer's professional comments and add the following to "3.1 Summary of the design thinking method" and “3.2.2 Results of the evaluation of the applicability of teaching” of the manuscript:

In this study, expert knowledge was utilized to assess the applicability of various design thinking methods in university education, and there were two stages in the process. In the first stage, to select a method that matches the curriculum of "locality" thematic design, a group of five teachers was organized to form a panel of experts. First, the leader of the focus group made a preliminary summary of more than 100 design thinking methods and organized them into a visual chart according to the six phases of design teaching (Fig 6), and then the leader of the group explained to the other members the characteristics of the methods as well as analyzed the process of their application, and then asked each member to express their understanding and opinion about each design thinking method, and finally all members discussed the advantages and disadvantages of each method and selected 71 methods suitable for the design course, which were summarized into 22 categories. 

In the second stage, to analyze the applicability of the selected 22 categories of methods in the dimensions of "design practice" and "locality", an evaluation team consisting of 36 experts was formed to evaluate the methods by the Delphi method, and the specific steps were as follows:

1, The team leader described and explained the characteristics and application process of the 22 methods to the members.

2, The opinions of the members were collected using questionnaire statistics, and each member could view the evaluation opinions of other members and encourage mutual discussion.

3, The questionnaires were all returned and valid, and finally, the assessment values of each member were integrated and analyzed.

The above will be added to the manuscript.

2.At the same time, it is worth considering whether uncertainty could have been modeled in a different way, perhaps more suitable for the analyzed problem. The authors should consider whether there are other methods of identifying expert knowledge that could have been used in this context, and whether the methodology employed, such as the RANCOM method, could have been more appropriate or simply more accessible to readers.

Response: Thanks to the reviewers' suggestions, we will add the following to "1.3 Selection of teaching methods" in the manuscript.

Evaluation information in the form of language is not directly involved in mathematical operations, and decision-making on it requires multiple experts to form a group decision, which then requires information transformation and aggregation of the information of each program, followed by sorting, comparison, and selection of the best. At present, the methods of processing linguistic evaluation information are mainly divided into three categories: one is the analytical method of evaluation information transformation, which transforms linguistic evaluation information into fuzzy numbers, and then carries out arithmetic operations and analysis, such as the fuzzy analytic hierarchy process (F-AHP) (Afolayan et al., 2020); the second is the symbol transfer-based analysis method, which is based on the calculation of linguistic evaluation information values directly from the set of linguistic evaluation information, such as RANking COMparison (RANCOM) (Wieckowski et al., 2023); the third category of methods is the 2-tuples linguistic analysis method (Malhotra & Gupta, 2023 ), which aims to transform the linguistic evaluation information given by the decision maker into 2-tuples linguistic symbols, and then use the 2-tuples linguistic algorithm and the agglomerative operator to perform information agglomerative analysis. By comparing and analyzing the above methods, it is found that the 2-tuples linguistic analysis method has great superiority in dealing with model and information loss, which makes the calculation results of linguistic evaluation information more accurate.

The following literature is added to the paper:

29.Wieckowski J, Kizielewicz B, Shekhovtsov A, Sałabun W. Rancom: A novel approach to identifying criteria relevance based on inaccuracy expert judgments. Engineering Applications of Artificial Intelligence. 2023; 122: 106114.

30.Afolayan AH, Ojokoh BA, Adetunmbi AO. Performance analysis of fuzzy analytic hierarchy process multi-criteria decision support models for contractor selection. Scientific African. 2020; 9: e00471.

31.Malhotra T, Gupta A. A systematic review of developments in the 2-tuple linguistic model and its applications in decision analysis. Soft Computing. 2023; 27(4): 1871-1905.

3.Furthermore, there is potential to increase attention to the sensitivity associated with the applied uncertainty generalization. The authors should consider the possibility of more detailed discussion of this issue to provide readers with a fuller understanding of the analyzed results.

Response: Thanks to the reviewer's suggestion, we will add the following content in "4.1 Selection of design thinking approaches in localized design courses" of the manuscript.

The actual assessment and decision-making problems are generally complex things, and the multi-criteria assessment, although it can synthesize multiple factors to analyze the problem, is still one-dimensional, and the mutual exclusion and influence between the criteria belongs to the continuous multi-dimensional structure. In addition, the ambiguity and uncertainty of human thinking, as well as the finite nature of individual knowledge, experience, and ability to solve the decision-making problems faced often do not have all the necessary knowledge and information if they rely only on individual experience and wisdom. Although groups of experts have certain common behavioral and professional characteristics at a particular time, such as cultural level and value orientation. However, the fact that these characteristics of expert groups are often abstract, the form of evaluation given to alternatives is uncertain linguistic information, and the difficulty for individuals to understand the full picture of the group and the sometimes one-sided understanding of the decision-making goals makes it difficult for individual decision-makers to accurately grasp the behavioral characteristics of the group. In this study, thematicity and practicability were two important dimensions in determining the appropriateness of the pedagogical methods used in the design course, and the two were not isolated, but rather in a state of interaction, which added to the complexity of the decision. When assessing the thematic dimension also affects the practicality, therefore, the method proposed in this study is a continuous linguistic representation of the evaluation information, which is more suitable for the 2 dimensions and avoids missing and distorted information. The preferred approach through visualization of graphs is more intuitive and precise than numerical. The main objective is to screen and develop appropriate design teaching methods by collecting group opinions from university teachers with rich design teaching experience, analyzing and evaluating them, and identifying the knowledge of the expert group.

---

## [Decision Letter · Decision Letter 2]

29 Feb 2024

How to choose a design thinking method for teaching the design of localization: A two-dimension linguistic fuzzy model with two-tuples

PONE-D-23-35366R2

Dear Dr. Fu,

We’re pleased to inform you that your manuscript has been judged scientifically suitable for publication and will be formally accepted for publication once it meets all outstanding technical requirements.

Kind regards,

Ta-Chung Chu

Academic Editor

PLOS ONE

Additional Editor Comments (optional):

Reviewers' comments:

Reviewer's Responses to Questions

**Comments to the Author**

1. If the authors have adequately addressed your comments raised in a previous round of review and you feel that this manuscript is now acceptable for publication, you may indicate that here to bypass the “Comments to the Author” section, enter your conflict of interest statement in the “Confidential to Editor” section, and submit your "Accept" recommendation.

Reviewer #1: All comments have been addressed

2. Is the manuscript technically sound, and do the data support the conclusions?

Reviewer #1: Yes

3. Has the statistical analysis been performed appropriately and rigorously? 

Reviewer #1: N/A

4. Have the authors made all data underlying the findings in their manuscript fully available?

Reviewer #1: Yes

5. Is the manuscript presented in an intelligible fashion and written in standard English?

Reviewer #1: Yes

6. Review Comments to the Author

Reviewer #1: The paper has undergone significant improvements and is now deemed suitable for acceptance in its present state.

7. PLOS authors have the option to publish the peer review history of their article (what does this mean?). If published, this will include your full peer review and any attached files.

Reviewer #1: No

---

## [Editor Report · Acceptance letter]

26 Mar 2024

PONE-D-23-35366R2 

PLOS ONE

Dear Dr. Fu, 

I'm pleased to inform you that your manuscript has been deemed suitable for publication in PLOS ONE. Congratulations! Your manuscript is now being handed over to our production team.

Kind regards, 

on behalf of

Dr. Ta-Chung Chu 

Academic Editor

PLOS ONE